  

# KANSL3 directs transcriptional programs essential for hepatic metabolism and differentiation

Meike Wiese[1,3] ⓘ, Cecilia Pessoa Rodrigues[1], Mehmet Eray Akbas[1,2,3], Yidan Sun[1], Herbert Holz[1] ⓘ, Juan Alfonso Martinez Greene[1,2], Tsz Hong Tsang[1] ⓘ, Chiara Bella[1] ⓘ, Kerstin Ganter[1], Thomas Stehle[1], Maria Shvedunova[1] ⓘ, Asifa Akhtar[1,3] ⓘ

**Liver disease is a leading cause of mortality worldwide. Emerging evidence highlights the significant role of epigenetic regulation in sustaining liver homeostasis, providing new therapeutic strategies for liver disease. Hepatocyte-specific deletion of the epigenetic regulator KANSL3, a key component of the NSL complex, results in early-onset liver disease marked by biliary hyperplasia and hepatic fibrosis. KANSL3 is essential for regulating hepatocyte transcriptional networks important for hepatic steroid and lipid metabolism through histone acetylation. Moreover, single-cell RNA sequencing demonstrated that the loss of KANSL3 disrupts the differentiation of hepatocytes in vivo. The transcriptional programs necessary for hepatocyte differentiation of ductal and fetal liver organoids were severely compromised in the absence of KANSL3. These findings collectively demonstrate a crucial role of the epigenetic regulator KANSL3 in hepatocyte differentiation in liver development and disease.**

## Introduction

Liver disease is a major health crisis, resulting in around two million deaths per year worldwide (Asrani et al, 2019). At late stages, liver disease is marked by cirrhosis and may increase the risk of developing hepatocellular carcinoma (HCC). The liver is known for its remarkable plasticity and regenerative ability (Gadd et al, 2020). Research of the last decades revealed the inherent adaptability of hepatic epithelial cell types, which include hepatocytes and biliary epithelial cells (BECs), suggesting that these cells may serve as facultative liver stem cells with the ability to replenish both lineages in response to different types of liver injury (Gadd et al, 2020). The transdifferentiation of hepatocytes into BECs is a process observed during severe bile duct injuries. In contrast, the conversion of BECs into hepatocytes typically occurs in the context of persistent damage to hepatocytes, for example, during chronic hepatitis. Although the mechanisms governing the BEC-to-hepatocyte transition are not fully understood, emerging evidence indicates that BECs may undergo an intermediate progenitor stage during dedifferentiation before ultimately differentiating into the hepatocyte lineage (Raven et al, 2017; Aloia et al, 2019; Pu et al, 2023b). The molecular mechanisms underlying these cell transitions, however, are just beginning to be understood.

Epigenetic mechanisms play a vital role in guiding cell-fate decisions in both embryonic and adult tissues. Evidence indicates that the adult epigenetic landscape is dynamically regulated, promoting cellular plasticity and fate changes in response to tissue injury (Rajagopal & Stanger, 2016). Increasingly, studies show that alterations in the chromatin landscape drive cell-fate transitions during liver regeneration (Aloia, 2021). For example, transitions to liver progenitor states are facilitated by active histone modifications that increase chromatin accessibility through ARID1A, a key component of the SWI/SNF complex (Li et al, 2019), or by the oxidation of 5-methylcytosine to 5-hydroxymethylcytosine mediated by TET1 (Aloia et al, 2019). Importantly, epigenetic regulation is a very dynamic process enabling liver epithelial cells to return to a homeostatic state after the resolution of injury, making it an attractive target for therapeutic intervention.

Histone acetylation is critical for maintaining chromatin structure, transcription, and cellular function and is intricately linked to metabolic processes. The nonspecific sex lethal (NSL) complex catalyzes H4 acetylation (H4K16ac, H4K8ac, and H4K5ac) (Cai et al, 2010; Tsang et al, 2023; Radzisheuskaya et al, 2021) and functions as a transcriptional activator by binding to gene promoters and facilitating the recruitment of RNA polymerase II (Raja et al, 2010; Feller et al, 2011; Lam et al, 2012; Chelmicki et al, 2014) and transcriptional activators such as BRD4 (Raja et al, 2010; Gaub et al, 2020). It plays a key role in regulating tissue- and cell-specific functions (Karoutas et al, 2019; Pessoa Rodrigues et al, 2020, 2021; Sheikh et al, 2020; Tsang et al, 2023).

[1]Department of Chromatin Regulation, Max Planck Institute of Immunobiology and Epigenetics, Freiburg, Germany    [2]Faculty of Biology, Albert Ludwigs University Freiburg, Freiburg, Germany    [3]CIBBS - Center for Integrative Biological Signalling Studies, University of Freiburg, Freiburg, Germany

Correspondence: akhtar@ie-freiburg.mpg.de

To investigate the role of the NSL complex in the liver, we focused on the subunit KANSL3, a key activator of NSL-mediated transcriptional activation (Raja et al, 2010; Gaub et al, 2020). We found that the deletion of KANSL3 in hepatocytes in vivo leads to early-onset liver disease, characterized by biliary hyperplasia and fibrosis. Our results also showed that KANSL3 regulates gene expression of pathways related to steroids and lipid metabolism, which are essential for proper liver function. We demonstrated that the presence of KANSL3 is crucial for activating transcriptional programs that promote the differentiation of both biliary cells and fetal liver cells into hepatocytes. These findings highlight the importance of NSL complex–mediated gene regulation in liver development and disease.

# Results

### Hepatocyte-specific deletion of the NSL complex causes liver disease

To investigate the tissue-specific role of the NSL complex in liver homeostasis, *Kansl3*[fl/fl] mice were bred with mice expressing Cre recombinase under the control of the *Alb* promoter (*Kansl3* LKO) (Fig 1A). *Alb* is expressed in epithelial cells in the liver, primarily in hepatocytes, but also in subtypes of BECs (Planas-Paz et al, 2019). Near-complete knockout of *Kansl3* in the liver (LKO) was validated through mRNA and protein analyses of bulk liver tissue (Figs 1B and S1A and B). The residual amounts of the KANSL3 protein and *Kansl3* mRNA observed in *Kansl3 LKO* bulk liver tissue likely originate from nonepithelial cell types including fibroblasts, endothelial cells, or Kupffer cells. Depletion of *Kansl3* also led to decreased mRNA expression of *Kat8* (MOF) and to a lesser extent *Kansl1*, two other components of the NSL complex (Fig S1A). At the protein level, however, all members of the NSL complex were decreased upon depletion of KANSL3 (Fig 1B), indicating that the integrity of the NSL complex in the liver depends on the presence of KANSL3. Although *Kansl3* LKO animals display normal lifespan and are fertile, their body weight was overall reduced with the most significant drop in body weight observed in 3-wk-old animals (Fig S1C and D, Table S1). *Kansl3* LKO mice also showed mildly increased liver weights with the most significant changes observed 6 wk after birth (Fig S1E, Table S1). The livers of 3-wk-old *Kansl3* LKO animals already showed notable morphological differences, including pale tissue color and rigidity with fibrotic strands (Fig 1C). Correspondingly, *Kansl3* loss led to decreased levels of albumin, elevated total and direct bilirubin levels, and elevated levels of aspartate aminotransferase (AST) and triglycerides (TGA) in the serum of 3-wk-old animals (P21) (Fig 1D–G).

To examine the mice for signs of metabolic dysfunction–associated fatty liver disease (MAFLD) and metabolic dysfunction–associated steatohepatitis (MASH), we performed histopathological analysis of livers from 3 wk-old *Kansl3* LKO mice and littermate controls using the MASH/MAFLD activity scoring system adapted for rodents (Liang et al, 2014). Typical features of classical MASH progression, such as micro- and macrovesicular steatosis, were absent in *Kansl3* LKO mice at P21

(Figs 1H and S1F). However, picrosirius red staining showed increased deposition of collagen fibers in portal and periportal, as well as perisinusoidal, areas with bridging fibrosis in *Kansl3* LKO mice (severity score 1.6) (Figs 1H and S1F). This was further validated by immunofluorescence using a collagen I and III antibody (Fig 1I). Elevated levels of fibrosis in *Kansl3* LKO livers were also accompanied by increased a-SMA staining (Fig 1J). Although a-SMA signal was only detected around the veins in controls, in *Kansl3* LKO cells we observed increased labeling throughout the parenchyma (Fig S1G). Furthermore, *Kansl3* LKO mice developed increased incidence and severity of biliary hyperplasia and a ductular reaction compared with control animals (Figs 1H and S1F), indicating severe liver damage. This was further corroborated by immunofluorescence staining for KRT19 (Fig 1K), as well as fluorescence-activated cell sorting (FACS) analysis for EpCAM+ cells, revealing higher numbers of BECs in *Kansl3* LKO livers compared with controls (Fig S1H).

We conclude that KANSL3 is an essential component of the NSL complex in liver tissue, and its loss in hepatocytes leads to liver dysfunction in 3-wk-old mice resulting in liver fibrosis and the development of a biliary hyperplasia (ductular reaction).

### KANSL3 regulates metabolic genes in the liver

Bulk RNA-seq on liver tissue obtained from 3-wk-old *Kansl3* LKO mice revealed extensive rewiring of the liver transcriptome with 3,582 up- and 3,148 down-regulated genes compared with age-matched controls (FDR < 0.05, no $\log_2$FC cutoff; Figs 2A and S2A and B, Table S2). Down-regulated genes were specifically enriched for fatty acid and steroid metabolism pathways, as well as xenobiotic response processes (FDR < 0.05, $\log_2$FC < −1; Figs 2B and C and S2C), suggesting that key transcriptional programs that drive core hepatic functions were disrupted in *Kansl3* LKO livers. Confirming the gene expression changes, we detected increased lipid accumulation in the livers of P21 *Kansl3* LKO animals (Fig 2D). Total cholesterol levels also increased in *Kansl3* LKO livers at P21, which were not yet detectable at P7, indicating a worsening of metabolic dysfunction over time (Fig 2E).

At the same time, we observed a significant number of up-regulated genes that were enriched in terms associated with tissue remodeling, wound healing, and extracellular matrix organization, mirroring the increased fibrosis observed in the liver tissues of *Kansl3* LKO mice (FDR < 0.05, $\log_2$FC > 1; Figs 2B and S2D). In line with this, we noted a significant up-regulation of established liver fibrosis markers (*Col3a1*, *Col1a1*, *Col4a2*, *Lox1*, *Timp1*, *Vcan*) alongside newly discovered profibrotic genes (*Aepb1*, *Prrx1*, and *Creb3l1*) (Wang et al, 2021), as well as increased expression of genes indicative of myofibroblast activation and proliferation (*Acta2*, *Des*, *Ctgf*, and *Cyr61*) (Fig S2E). In addition, we noted up-regulation of proinflammatory response (*Ccl2*, *Tnf*) and monocyte infiltration genes (*Csfr1*, *Ccr2*, *Itgam*, *Ly6c1*) (Fig S2E) and a strong increase in biliary epithelial markers (*Sox9*, *Spp1*, *Epcam*, *Krt19*, *Cftr*) further confirming a ductular reaction (Fig S2E).

KANSL3 ChIP-seq analysis in control liver tissue revealed strong KANSL3 enrichment at the transcription start sites (TSS) of down-regulated genes in *Kansl3* LKO livers, whereas up-regulated genes showed weaker binding at only a subset of genes (FDR < 0.05,

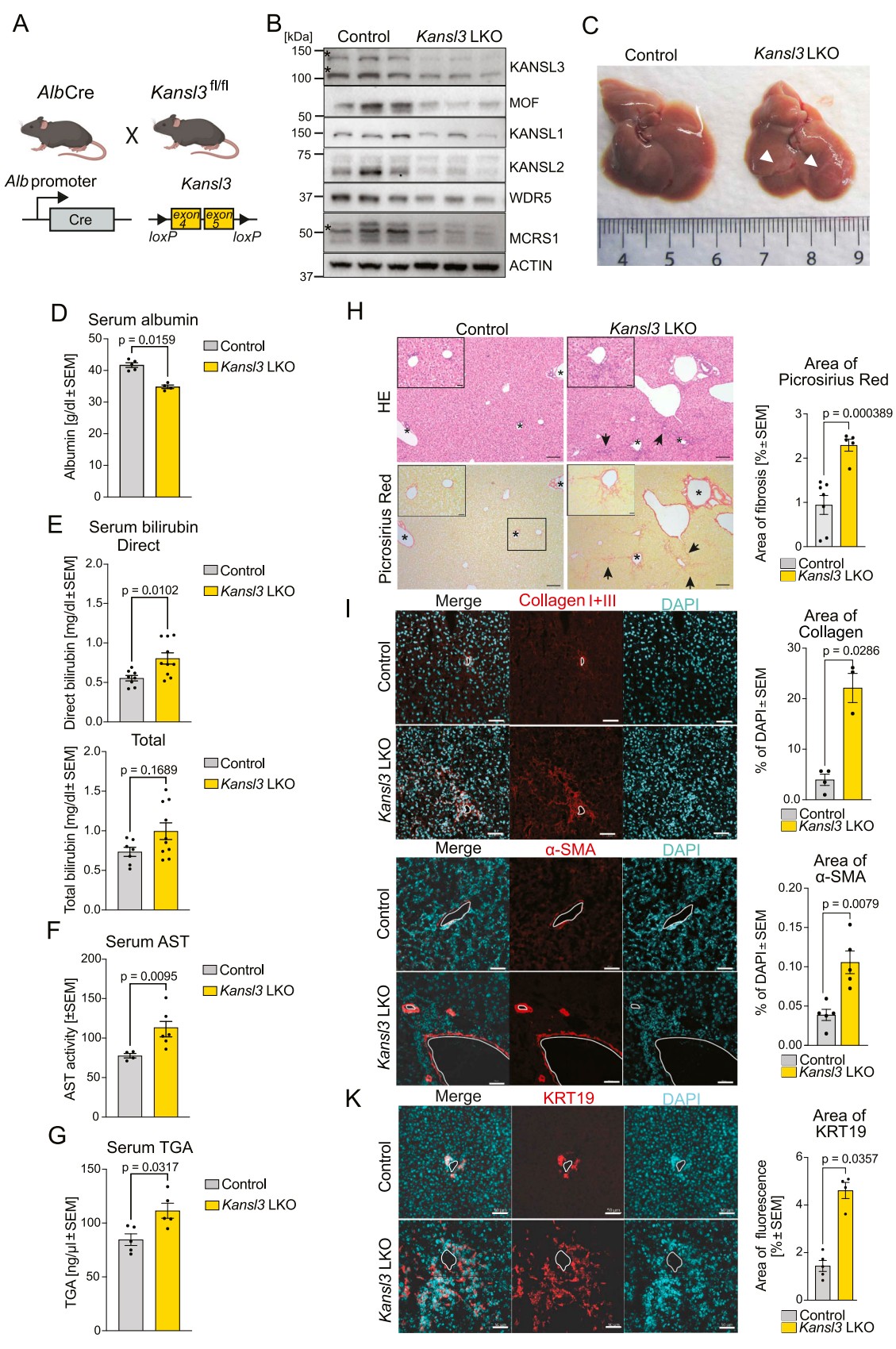

log$_2$FC > 1 or <−1; Fig 2F). Notably, KANSL3 was enriched at the TSS of genes related to fatty acid metabolism such as *Mecr* and *Acms2*, and steroid metabolism such as *Srebf1* and *Hsd11b1* (Fig 2G), as well as genes related to the response to xenobiotic stimuli such as *Xbp1* and *Blmh* (Fig S2F and G). At the same time, at up-regulated genes, for example, related to ECM organization or response to wound healing, KANSL3 was not enriched compared with a set of random genes (Fig S2F), indicating a secondary response to the loss of KANSL3 in hepatocytes in the tissue microenvironment.

Our analysis revealed significant rewiring of the liver transcriptome in *Kansl3* LKO mice, characterized by the down-regulation of key hepatocyte metabolic genes, many of which are directly regulated by KANSL3.

### KANSL3 regulates hepatic metabolic genes via histone acetylation

To study the mechanisms of transcriptional regulation by KANSL3 and at the same time overcome the limitations of bulk tissue analysis of mixed cell populations, we developed an in vitro hepatocyte model. By employing the degradation tag (dTAG) system in the murine liver cell line Hepa1-6, we examined the consequences of acute KANSL3 depletion. This approach enabled us to track the early transcriptomic changes in hepatocytes after acute KANSL3 loss. The dTAG system allowed for the complete degradation of the KANSL3 protein within 45 min after the addition of the dTAG-13 ligand (Fig S3A). RNA-seq analysis in Hepa1–6 HA-dTAG-KANSL3 cells after 24 hrs of KANSL3 degradation resulted in a drastic deregulation of the transcriptome, with 2,202 genes being down- and 1,762 up-regulated compared with cells treated with a control ligand (dTAG-NEG) (FDR < 0,05, not log$_2$FC cutoff; Fig S3B, Table S2). The down-regulated genes were enriched for mitochondrial functions, lipid metabolism, and cilium-related terms (Fig S3C and D), suggesting functional parallels between an acute depletion of KANSL3 in cells and the constitutive loss in liver tissue (Fig 2B). Notably, the expression of down-regulation was rescued by the ectopic expression of a FLAG-tagged human KANSL3 full-length protein (Fig S3E–G, Table S2), indicating direct regulation by KANSL3.

Next, by KANSL3 ChIP-qPCR using an anti-HA antibody, we confirmed specific binding of KANSL3 at the promoters rather than the 3′-ends of the target genes *Mrp11* and *Fah*, whereas

KANSL3 binding was abolished after dTAG-13 treatment (Fig S3H). ChIP-seq analysis demonstrated a significant reduction of the H4K16ac levels at KANSL3 target genes upon KANSL3 degradation compared with a set of random genes (Fig S3I). Furthermore, treatment of dTAG-13–treated cells with the HDAC inhibitor trichostatin A (TSA) rescued the gene expression of KANSL3 target genes *Mrp11*, *Fah*, *Mrp55*, and *Nduf3b* (Fig S3J) supporting acetylation-dependent gene regulation by KANSL3.

We conclude that KANSL3 is an essential component of the NSL complex, crucially modulating gene expression programs necessary for core hepatocyte functions through histone acetylation. Transcriptomic alterations induced by acute KANSL3 loss in hepatocytes closely resemble those observed with long-term KANSL3 depletion in vivo, establishing KANSL3 as a key regulator of hepatocyte gene expression.

### Old *Kansl3* LKO mice develop premalignant liver lesions

Considering the profound impact on liver health observed in 3-wk-old *Kansl3* LKO mice, next we investigated older animals to evaluate the long-term consequences of KANSL3 loss. Old *Kansl3* LKO mice (90-wk-old) exhibited a normal lifespan and no apparent physiological abnormalities. In fact, the elevated levels of liver disease markers observed in the serum of 3-wk-old *Kansl3* LKO animals improved with age (compare Figs 1D–G, S1F with Fig S4A). Although the histopathological changes observed in young mice persisted in older *Kansl3* LKO mice, they exhibited significantly reduced severity (Fig S4B), suggesting the development of compensatory mechanisms in the livers of *Kansl3* LKO mice over time. Despite the overall improvement of liver health, however, old *Kansl3* LKO mice developed multiple atypical lesions per liver compared with age-matched controls (Fig S4C). Histological analysis of control and *Kansl3* LKO liver tissue revealed increased proliferation, nuclear abnormalities along with the disruption of the acinar lobular architecture, and the appearance of "pseudorosettes" as hallmarks of hepatic premalignant growth (International Consensus Group for Hepatocellular Neoplasia, 2009) (Fig S4D and E).

Taken together, our findings showed that KANSL3 loss in hepatic epithelial cells leads to severe liver disease in young mice. Although the liver disease markers improve with age, long term, the tissue develops signs of premalignant growth.

---

**Figure 1. Hepatocyte-specific deletion of the NSL complex causes liver dysfunction.**
**(A)** *Kansl3*$^{fl/fl}$ mice were crossed to albumin Cre mice to generate hepatocyte-specific *Kansl3* LKO mice. Either *Kansl3*$^{fl/fl}$ or *Alb*Cre-positive (*Alb*Cre) mice serve as control groups in indicated experiments. **(B)** Western blot analysis of NSL complex members in whole-cell extracts of bulk liver tissue of 3-wk-old control (*Kansl3*$^{fl/fl}$, n = 3) and *Kansl3* LKO mice (n = 3). Actin served as a loading control. Asterisk indicates specific bands. **(C)** Representative images of liver from 3-wk-old control (*Alb*Cre) and *Kansl3* LKO mice. Arrowheads indicate fibrotic streaks in the tissue. **(D, E, F, G)** Serum albumin (control, n = 5; *Kansl3* LKO, n = 4), direct serum bilirubin (Control, n = 8; *Kansl3* LKO, n = 10), serum aspartate aminotransferase (AST) (control, n = 4; *Kansl3* LKO, n = 6), and triglyceride (TGA, control, n = 5; *Kansl3* LKO, n = 5) levels in 3-wk-old (P21) control (*Kansl3*$^{fl/fl}$) and *Kansl3* LKO mice. Significance was determined by the Mann–Whitney test; exact *P*-values are indicated in panels. **(H)** Left: representative images of HE (top) and Picrosirius red stainings (bottom) of livers from 3-wk-old (P21) control (*Kansl3*$^{fl/fl}$) and *Kansl3* LKO mice. Asterisks indicate increased biliary cellularity (HE) and increased collagen fibers (Picrosirius red) in portal and periportal areas, and arrows indicate increased cellularity in adjacent hepatocellular parenchyma (HE) or areas of bridging fibrosis (Picrosirius red). Scale bar: 100 *μ*m; scale bar zoom: 20 *μ*m. Right: Quantification of picrosirius red staining area from P21 control (*Kansl3*$^{fl/f}$, n = 4) and *Kansl3* LKO mice (n = 5). Significance was determined by ordinary two-way ANOVA; exact *P*-values are indicated in the figure. **(I, J, K)** Left: representative immunofluorescence images of liver sections from P21 control (*Kansl3*$^{fl/f}$) and *Kansl3* LKO mice using an anti-collagen I + III (I), anti-a-SMA (J), and anti-KRT19 (K) antibody (red). Nuclei were stained using DAPI (cyan). Right: quantification of immunofluorescence area of collagen (control, n = 4; *Kansl3* LKO, n = 3), a-SMA (control, n = 5; *Kansl3* LKO, n = 5), and KRT19 (control, n = 5; *Kansl3* LKO, n = 4). Significance was determined by the Mann–Whitney test; exact *P*-values are indicated in the figure. Scale bar: 50 *μ*m.
Source data are available for this figure.

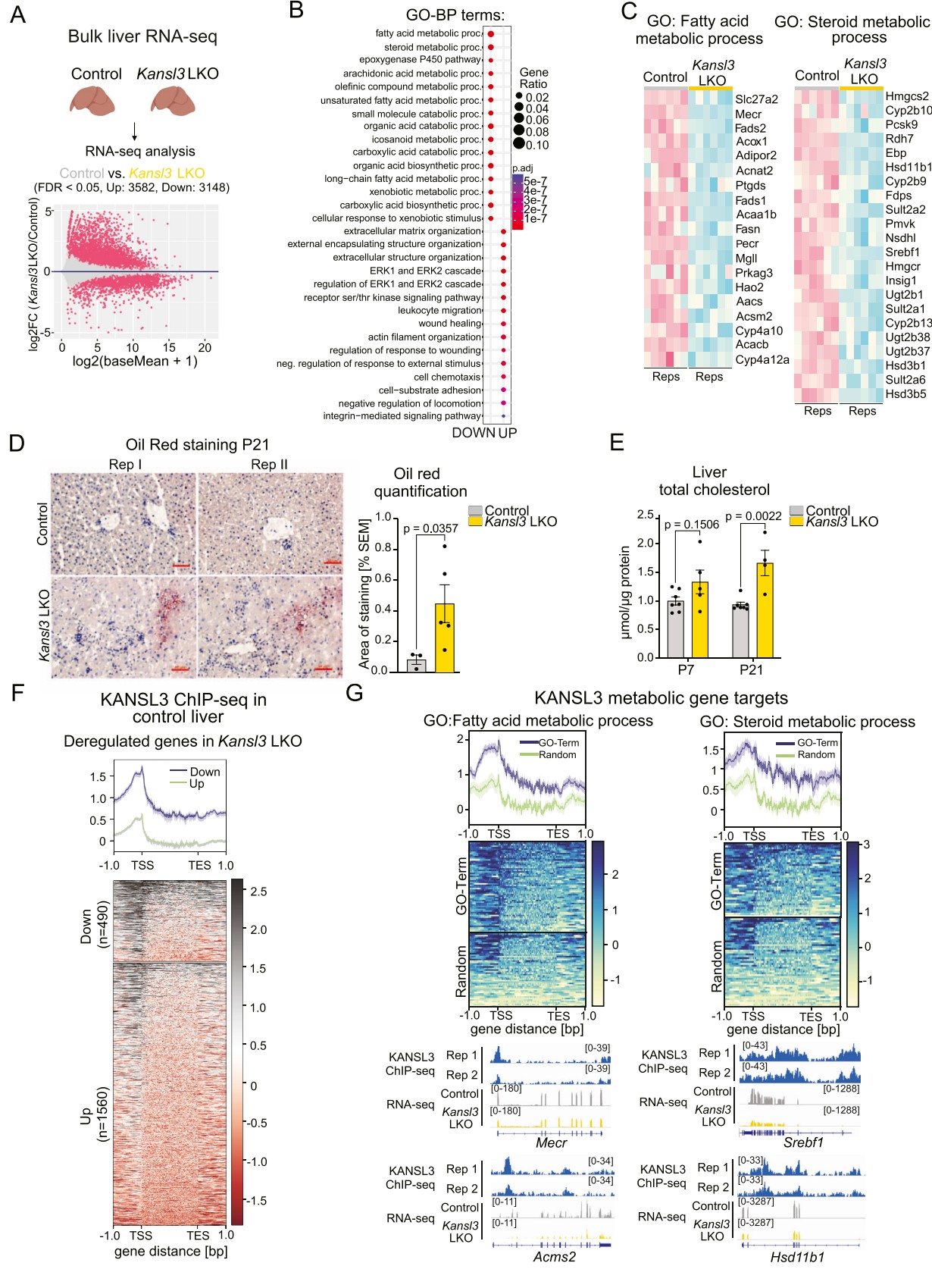

## Hepatic *Kansl3* loss causes imbalance of hepatocytes and BECs in neonates

Given the advanced disease phenotype observed in 3-wk-old mice, we decided to examine earlier developmental stages to capture the initial changes occurring in the livers of *Kansl3* LKO mice. *Alb*Cre transgene expression correlates with hepatic specification in the fetal liver showing mosaic activity from E15.5 onward and achieving recombination in ~80% of the cells at neonatal age P6 (Weisend et al, 2009). We performed histopathological analysis of livers at E17.5 and P7 (1 wk). Although the E17.5 fetal liver revealed normal hepatic development predominantly characterized by hematopoietic cells with scattered hepatic cell patches, there were no apparent differences in the histopathological findings between *Kansl3* LKO and age-matched controls at E17.5 (Fig 3A). P7 neonates of both genotypes showed residual foci of extramedullary hematopoiesis, as expected (Fig 3A). Remarkably, already in neonates *Kansl3* LKO mice developed fibrosis, a ductular reaction (Figs 3A and B and S5A and B), and serum analysis identified liver disease markers in *Kansl3* LKO neonates compared with age-matched controls (Fig S5C and D). These results indicate that loss of KANSL3 leads to developmental defects in the liver.

Next, we assessed the molecular changes in the liver microenvironment upon *Kansl3* deletion in hepatocytes using scRNA-seq of isolated cell populations enriched for BECs, stromal cells, and hepatocytes (Figs 3C and S5E and F) in neonatal mice. We analyzed a total of 6,145 cells from control samples, with a median of 2,456 genes per cell, and 4,886 cells from *Kansl3* LKO samples, with 3,120 genes per cell (Fig S5G and H). This approach provided a comprehensive dataset encompassing the primary liver cell types (Figs 3D and S5I, Table S3).

Analysis of cell composition revealed an imbalance in the numbers of hepatocytes (clusters 0 and 6) compared with BECs (cluster 1) in *Kansl3* LKO livers (Figs 3E and F and S5J). Hepatocyte cluster 0 contained mature hepatocytes characterized by the expression of key hepatocyte pathways involved in lipid, steroid, and amino acid metabolism (Fig S5K), and cluster 6 included hepatocytes that also displayed *Sox9* and *Epcam* positivity (Fig S5I). Notably, both clusters showed significantly reduced cell numbers in *Kansl3* LKO livers (Fig 3E). Conversely, the BEC compartment displayed an increase in cell numbers in *Kansl3* LKO livers compared with controls (Fig 3F). Cells within this cluster predominantly expressed genes associated with increased transcription, RNA processing, and the generation of

biomass consistent with enhanced proliferation (Fig S5L). The proliferative state of BECs was further supported by the expression of *Myc* and *Pcna* (Fig S5M). Collectively, these results corroborate with the ductular reaction observed in neonatal *Kansl3* LKO mice.

To validate the imbalance between hepatocytes and BECs, we crossed *Kansl3* LKO or *Alb*Cre control mice with mTmG reporter mice (Figs 3G and S5N). Liver cells from these mice were stained with an antibody against the BEC marker EpCAM (Fig 3G). FACS analysis confirmed an increase in the total number of EpCAM+ BECs together with a reduction of GFP+/EpCAM− hepatocytes (Fig S5O and P). The total BEC population was composed of GFP+ (~65%) and GFP− BECs (~35%), consistent with the heterogenic expression of *Alb* within the BEC compartment (compare Fig 3E). To understand the reason behind the imbalance, we performed pseudobulk differential expression analysis on hepatocytes (cluster 0) and BECs (cluster 1) (Table S4). In *Kansl3* LKO hepatocytes, we observed a down-regulation of genes related to mitochondrial function, ciliary landscape, fatty acid metabolism, and response to toxic substances, consistent with impaired hepatocyte function (Fig 3H, left). Up-regulated genes were strongly associated with apoptosis, ferroptosis, and stress signaling, indicating a decrease in viability of hepatocytes in *Kansl3* LKO neonates. This was further confirmed by Annexin V staining, which showed an increase in apoptotic GFP+ hepatocytes in mTmG *Kansl3* LKO animals compared with controls (Fig 3I). In contrast, BECs showed an up-regulation of cell–cell adhesion, proliferation, and VEGFA signaling, consistent with a ductular response (Fig 3H, right). Down-regulated genes were enriched for similar pathways as in hepatocytes (Fig 3H, right), which may reflect the heterogeneity within the BEC compartment during a ductular reaction.

Given the fibrosis phenotype observed in neonatal *Kansl3* LKO animals, we investigated stromal cell populations in clusters 3, 8, 10, and 12 (Fig 3D). Although all populations express hepatic stellate cell (HSC) markers *Acta2* and *Des*, certain differences across clusters were evident (Fig S6A).

Cluster 3 cells expressed high levels of fibroblast markers *Dcn* and *Dpt*, with a heterogeneous gene expression profile enriched for extracellular matrix organization and tissue morphogenesis (Fig S6A and B). Cluster 8 and 10 cells expressed high levels of *Acta2*, with cluster 8 cells in addition expressing vascular smooth muscle cell (VSMC) markers *Myh11* and *Cnn1*, and cluster 10 cells showing a more active HSC or myofibroblast signature expressing high levels of *Lox* and *Col1a1/2* and *Col3a1* (Fig S6A and C). Cluster

---

**Figure 2. KANSL3 regulates metabolic genes in the liver.**
**(A)** MA-plot showing bulk RNA-seq results (FDR: 0.05) of liver tissue of P21 control (*Alb*Cre, n = 6) and *Kansl3* LKO mice (n = 6). Genes with $log_2$FC < −0.5 or >0.5 are highlighted in red. **(B)** Gene ontology analysis of biological processes for deregulated genes (FDR < 0.05, $log_2$FC < −1 or >1) detected in P21 *Kansl3* LKO liver tissue. Color code indicates the adjusted *P*-value results, and the size of dots indicates the gene ratio per GO term. The top 15 GO terms for both down- and up-regulated genes are displayed. **(C)** Heatmap displaying the gene expression of down-regulated genes in *Kansl3* LKO liver tissue (FDR < 0.05, $log_2$FC < −1) for the GO-term fatty acid metabolic process and steroid metabolic process. Z-scores of normalized counts are indicated. **(D)** Left: representative images of Oil Red O staining of livers from 3-wk-old (P21) control (*Kansl3*$^{fl/fl}$) and *Kansl3* LKO mice. Nuclei are counterstained with hematoxylin. Right: quantification of the Oil Red O staining in *Kansl3*$^{fl/fl}$ (n = 3) and *Kansl3* LKO mice (n = 3). Significance was determined by the Mann–Whitney test; exact *P*-value is indicated in the figure. Scale bar: 50 *μ*m. **(E)** Quantification of the total cholesterol in livers from neonatal (P7) and 3-wk-old (P21) control (*Kansl3*$^{fl/fl}$ n = 7) and *Kansl3* LKO (P7, n = 5; P21, n = 4) mice. Cholesterol concentrations are normalized to total liver protein and are in addition normalized to control samples. Significance was determined by the Mann–Whitney test, and the exact *P*-value is indicated in the figure. **(F)** KANSL3 ChIP-seq levels at all deregulated genes (FDR < 0.05, $log_2$FC < −1 or $log_2$FC > 1) in *Kansl3* LKO P21 livers. Blue lines indicate down- and green lines up-regulated genes. **(G)** Top: KANSL3 ChIP-seq levels in control livers at down-regulated genes (FDR < 0.05, $log_2$FC < −1) of indicated GO terms (blue lines) compared with a set of random genes (green lines). Bottom: RNA-seq and KANSL3 ChIP-seq tracks of representative KANSL3 target genes of the GO-term fatty acid metabolic process (*Mecr* and *Acms2*) and steroid metabolic processes (*Srebf1* and *Hsd11b1*) in control and *Kansl3* LKO liver tissue.

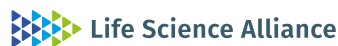

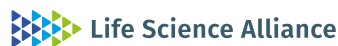

12 cells showed overall lower levels of active HSC markers (Fig S6A and C).

In terms of cell numbers, clusters 3 and 10 showed a reduction in *Kansl3* LKO mice (Fig S6D), whereas cluster 8 showed a mild increase, and cluster 12 showed a more pronounced increase. Notably, pseudobulk differential gene expression analysis between control and *Kansl3* LKO mice confirmed an up-regulation of pathways related to fibrosis and tissue reorganization, such as collagen catabolism, keratan sulfate degradation, or disease of glycosylation (Fig S6E), consistent with the increased fibrosis observed in *Kansl3* LKO animals.

In summary, our data indicate that the disease phenotype of *Kansl3* LKO mice is already established in neonates. scRNA-seq analysis of neonatal control and *Kansl3* LKO livers uncovered a significant shift in liver cellular composition. *Kansl3* LKO animals exhibited a notable deficiency of functional hepatocytes, which was partly linked to an increase in apoptosis. At the same time, numbers of BECs were markedly increased consistent with a ductular response phenotype and an expression profile indicative of a fibrogenic state in stromal cells.

### Deletion of *Kansl3* impairs hepatocyte differentiation in mice

Although *Kansl3* LKO animals exhibited increased hepatocyte apoptosis, the presence of viable, albeit functionally impaired, hepatocytes suggests that the observed imbalance is not solely due to reduced hepatocyte viability. BECs have been shown to have the potential to transdifferentiate into functional hepatocytes in response to chronic or severe acute hepatocyte damage (Español-Suñer et al, 2012; Raven et al, 2017; Deng et al, 2018). Given the strong evidence for a ductular reaction, we hypothesized that BECs encountered difficulties in differentiating into functional hepatocytes. To investigate this, we extracted the liver epithelial cells from our scRNA-seq data (highlighted in Fig 3D by a dashed line). Reclustering of the epithelial cell subset revealed a strong heterogeneity within both the BEC (C0, C4, C5, C10, C12) and hepatocyte (C1, C2, C3, C6) compartments. Intermediate clusters expressing markers for both cell types were also identified (C7, C3) (Figs 4A–C and S6F, Table S3). Notably, BEC cluster C5 showed an inflammatory signature (*Ccl2, Cxcl1, Cxcl2,* and *Tnf* expression), whereas cluster C10 exhibited a more proliferative profile (*Pcna, Mki67, Cks1b,* and

*Cdc20* expression) (Planas-Paz et al, 2019). Clusters C0, C4, and C12 expressed markers of BEC progenitors (*Cd24, Prom1, St14*) (Lu et al, 2015; Li et al, 2017) (Table S3). Interestingly, an expression gradient of the BEC marker *Tspan8* was detected across all BECs, with the highest expression in C12 and C4, medium expression in C0 and C10, and low expression in cluster C5 (Fig S6F). Hepatocyte clusters C1, C2, and C6 all express high levels of core hepatocyte metabolic genes, whereas C6 also showed a proliferative signature (*Cdc20, Cdk1, Cdkn3*) (Fig 4B, Table S3). Remarkably, *Kansl3* LKO mice showed a significant reduction in hepatocyte clusters C1 and C6, and an almost complete absence of the intermediate clusters C7 and C3 (Fig 4D, Table S3), whereas cluster C2 showed equal numbers (Fig 4D, Table S3). Within the BEC compartment, an increase in clusters C0, C4, and C10 was observed, accompanied by a decrease in cluster C5 (Fig 4D, Table S3).

To determine whether the cell fate toward the hepatocyte lineage was affected in *Kansl3* LKO animals, we used FateID (Herman & Sagar, 2018). This iterative supervised learning algorithm calculates the probability of a cell's fate trajectory, tracing mature cell types back to their potential progenitor states. We designated cluster 3 as the terminally differentiated state of hepatocytes based on high expression levels of hepatocyte markers (Figs 4C and E and S6F and G) and cluster 9 as the terminally differentiated state of HSC identified by the expression of HSC marker genes (Figs 4E and S6G and H) serving as the algorithm's anchor point. We ordered 875 control and 1,444 *Kansl3* LKO cells and analyzed the expression of 728 genes along the hepatocyte differentiation trajectory. Genes with similar expression patterns were grouped into nodes (Pearson's correlation coefficient >0.85) (Fig 4F and G). Although BEC lineage–specific genes were primarily expressed in nodes 1–4, hepatocyte lineage–specific genes were predominantly found in nodes 9 and 10, with nodes 7 and 8 representing intermediate states. We observed an accumulation of *Kansl3* LKO cells expressing BEC lineage–specific genes (e.g., node 1), while cells expressing hepatocyte lineage–specific genes were markedly depleted in *Kansl3* LKO samples (e.g., node 10) (Fig 4F). Importantly, fate-bias analyses using FateID revealed that *Kansl3* LKO cells had a reduced probability of differentiating into terminally differentiated hepatocytes (Fig 4G–I), consistent with the observed reduction in hepatocyte numbers in *Kansl3* LKO animals (Fig 3E). Overall, these findings indicate that *Kansl3* LKO hepatic

---

**Figure 3. Hepatic KANSL3 loss causes imbalance of hepatocytes and BECs in neonates.**
**(A, B)** Representative pictures of HE (A) and Picrosirius red stainings (B) of livers from E17.5 embryos and neonatal (P7) control (*Kansl3*fl/fl) and *Kansl3* LKO mice. **(A, B)** P7: asterisks indicate increased biliary cellularity (A) or increased collagen fibers (B) of portal and periportal areas, and arrows indicate foci of extramedullary hematopoiesis. Scale bar: 100 μm; scale bar zoom: 20 μm. **(C)** Schematic of experimental approach to investigate both the effects of *Kansl3* loss in hepatocytes on the tissue microenvironment in young animals using scRNA-seq in liver cells isolated from neonatal control (*Alb*Cre) and *Kansl3* LKO mice. The right panel depicts the flow cytometry gating strategy used to sort the sequenced cells. **(D)** Uniform Manifold Approximation and Projection (UMAP) clustering of liver cells from neonatal control (*Alb*Cre) and *Kansl3* LKO mice. Colors correspond to annotated cell types. Each dot represents a single cell, n = 3 animals per genotype. Marker genes per cluster are summarized in Table S3. **(E, F)** Left: number of control (gray) or *Kansl3* LKO (yellow) cells in hepatocyte (left panels, clusters 0 and 6) (E) or BEC clusters (right panel, cluster 1) (F). Enrichment was calculated by Fisher's exact test, and *P*-values are depicted in the figure. **(E, F)** Right: UMAPs for (E) hepatocyte marker genes *Alb* and *Asgr1* and (F) BEC marker genes *Epcam* and *Krt19*. **(G)** Top: *Kansl3*fl/fl mice were crossed to *Alb*Cre mice to generate hepatocyte-specific *Kansl3* LKO mice. Mice were further crossed with mTmG reporter mice to genetically label *Alb*-expressing cells. *Alb*Cre+ mTmG mice served as a control. Bottom: single-cell suspensions of P7 livers were stained with APC-EpCAM antibodies and analyzed by FACS analysis to dissociate EpCAM−/GFP+ hepatocytes from EpCAM+ BECs. **(H)** Gene ontology analysis of differentially regulated genes (FDR < 0.05) in *Kansl3* LKO animals for the hepatocyte cluster 0 (left) and the BEC cluster 6 (right) obtained by pseudobulk differential expression analysis. Color scheme indicates −log (*Q*-value). White color indicates a −log (*Q*-value) <−30. The complete list of deregulated genes is summarized in Table S4. **(I)** Representative FACS plots of Annexin V staining of EpCAM−/GFP+ hepatocytes from mTmG control and *Kansl3* LKO mice. Quantification of Annexin V-BV421+ (apoptotic) hepatocytes depicted as a percentage of viable (Zombie NIR-) hepatocytes in mTmG controls (n = 7) and *Kansl3* LKO mice (n = 8). Significance was determined by the Mann–Whitney test; exact *P*-value is indicated in the figure.

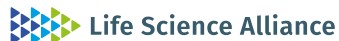

epithelial cells may be impaired in their ability to differentiate into mature hepatocytes.

### KANSL3 guides hepatocyte differentiation of ductular organoids

To study the role of KANSL3 in the differentiation of BEC progenitors into hepatocytes, we employed a hepatic organoid differentiation system, which allowed us to track dynamic changes in real-time generated ductal organoids from *Kansl3*$^{fl/fl}$ (iWT) and *Kansl3*$^{fl/fl}$ *Cag-Cre-ERT2*$^{T/+}$ (*Kansl3*-iKO) neonatal mice allowing for Cre-mediated recombination to be induced by 4-hydroxytamoxifen treatment (4OHT) (Fig 5A). Based on our scRNA-seq analysis, we developed a sorting strategy to enrich putative progenitor cells within the BEC compartment, as predicted by FateID: progenitor cells were characterized by their co-expression of *Epcam* (Fig 4D), *Prom1*, *CD24*, and *St14* (Fig S7A). In addition, we observed a marked enrichment of *Tspan8* expression in these cells (Fig S7A). Putative progenitor cells were sorted from neonatal mice based on the high expression of cell surface markers EpCAM, TSPAN8, and CD24, and used to generate bipotent ductular organoids (Fig S7B). WT EpCAM$^{high}$/TSPAN8$^{high}$/CD24$^+$ cells showed higher organoid-forming potential compared to those with lower *Tspan8* expression (EpCAM$^{high}$/TSPAN8$^{medium}$ or EpCAM$^{high}$/TSPAN8$^{low}$) (Fig S7C). Notably, the same cell populations isolated from neonatal *Kansl3* LKO mice showed reduced organoid formation capacity in vitro (Fig S7C), suggesting impaired self-renewal potential of bipotent progenitors in *Kansl3* LKO mice. To assess the impact of *Kansl3* depletion on hepatocyte differentiation in organoids, we cultured iWT and *Kansl3*-iKO ductal organoids in differentiation media for 12 d to induce hepatocyte differentiation (BEC-to-Hep), following a standard protocol (Huch et al, 2015) (Fig 5A). Hepatocyte differentiation was confirmed by qRT–PCR showing a gradual increase in the expression of hepatocyte markers *Alb* and *Cyp2d10* and a decrease in the ductular marker *Spp1* (Fig S7D). 4OHT was applied during the initial 48 h to induce recombination, and successful KANSL3 depletion was confirmed at both the protein and mRNA levels (Fig S7E–G). On the final day of differentiation, KANSL3 depletion led to notable morphological changes in hepatic organoids, which appeared smaller and exhibited reduced browning compared with controls (Fig 5B). A proliferation assay performed over the course of differentiation

confirmed a significant decrease in cellular proliferation in *Kansl3* iKO BEC-to-Hep organoids after 4OHT treatment on day 3 (Fig 5C) with no significant evidence for an increase in apoptosis at day 15 (Fig 5D).

We conducted bulk RNA-seq analysis at multiple time points (days 3, 4, 6, 8, and 15) upon KANSL3 depletion to follow temporal changes in gene expression. Differentially expressed genes revealed distinct patterns of up- or down-regulation in *Kansl3*-iKO BEC-to-Hep organoids over time (Fig S8A, Table S5). Overall, gene expression changes because of KANSL3 loss emerged as early as day 6 of hepatocyte differentiation and continued to evolve over time (Figs 5E and S8A). Genes typically activated during differentiation in iWT organoids were not properly up-regulated in *Kansl3*-iKO organoids (Fig 5E), and we further validated these results by qRT–PCR (Fig S8B), indicating that the presence of KANSL3 is important for proper activation of genes during hepatocyte differentiation. In addition, genes normally repressed during hepatocyte differentiation were not adequately repressed (Fig S8C).

Focusing on liver-specific genes comprised from various mouse datasets (Planas-Paz et al, 2019; Ardisasmita et al, 2022; Pu et al, 2023a) (Table S6), we observed that down-regulated genes in *Kansl3*-iKO Hep-to-BEC organoids were enriched for hepatocyte-specific functions, including steroid, lipid, and xenobiotic metabolism (Fig 5F–H). Similar pathways were identified in bulk liver tissue from *Kansl3* LKO mice (Fig 2B), indicating important parallels between the in vivo data and the in vitro results obtained from the organoids. At the same time, we observed a retention of biliary genes such as *Sox9* and *Spp1* (Fig 5G).

Ultimately, KANSL3 ChIP-seq data in control liver tissue confirmed an enrichment of KANSL3 at the TSS of hepatocyte-specific genes that were not expressed in ductal organoids at day 3 and were inadequately activated in *Kansl3*-iKO organoids at day 15 of differentiation such as *Cyp3a13*, *Nr1i2*, *Ugt2a3* (Fig 5I), and others (Fig S9), implying that the presence of KANSL3 is indispensable for the activation of transcriptional programs promoting hepatocyte differentiation.

Together, highly consistent with our in vivo findings, the BEC-to-Hep organoid data emphasize the crucial role of KANSL3 in the transcriptional activation of genes important for hepatocyte differentiation.

---

**Figure 4. Deletion of KANSL3 impairs hepatocyte differentiation in mice.**
**(A)** UMAP clustering of a subset of hepatic epithelial cells from neonatal control (*Alb*Cre) and *Kansl3* LKO livers. Marker genes per cluster are summarized in Table S3. **(B, C)** UMAPs for (B) hepatocyte marker genes *Alb*, *Asgr1*, *Hnf4a*, and *Cyp3a11* or (C) BEC marker genes *Epcam*, *Krt19*, *Hnfb1*, and *Sox9*. **(D)** UMAPs of cells from control (left) and *Kansl3* LKO (right) livers. The cell composition per cluster is summarized in Table S3. **(E)** UMAP based on RaceID clustering of hepatic epithelial cell subset. FateID was used to determine the cell fate toward the hepatocyte lineage in control and *Kansl3* LKO cells. The FateID algorithm calculates the probability of a cell's cell-fate trajectory, tracing mature cell types back to their potential progenitor states. Cluster 3 (hepatocytes, circle) and cluster 9 (HSCs, dashed circle) were set as the terminally differentiated states that were determined by the high expression of the marker genes, for example, *Alb*, *Asgr1* (hepatocytes), and *Acta2* and *Des* (HSCs), and further confirmed by low transcription entropy levels for both target clusters. The pseudo-order of the differentiation trajectory was manually added by a black line based on FateID prediction shown in Fig S6G. **(F)** Self-organizing map (SOM) generated from the hepatocyte trajectory (left). For SOM ordering, 875 control and 1,444 *Kansl3* LKO cells were extracted from the principal curve and fitted to cells with a fate bias of >0.4. The SOM shows the cells in pseudotemporal order on the x axis and transcripts that have similar expression profiles along the trajectory aggregated into nodes on the y axis (Pearson's correlation coefficient, >0.85), where eventually, every transcript corresponds to a line. The expression of 728 genes along the hepatocyte differentiation trajectory was analyzed. Right panels show normalized expression of all genes belonging to SOM nodes 10 and 1 in individual control (gray) or *Kansl3* LKO (yellow) cells along the hepatocyte differentiation trajectory. **(G)** Left panels: normalized expression of *Alb* and *Sox9* in cells along the hepatocyte differentiation trajectory predicted by FateID. Cells are colored according to the RaceID clusters in (E). Right panels: normalized expression of *Alb* and *Sox9* plotted against each cell's hepatocyte fate probability. **(H)** UMAP showing the hepatocyte fate-bias score. Fate bias was calculated per cell (dot in the UMAP), and colors indicate the power of the bias, 1 being the highest (red) and 0 the lowest (blue) probability of becoming a hepatocyte. **(I)** Hepatocyte fate bias of control (gray) and *Kansl3* LKO (yellow) cells as determined by FateID. Significance was determined by the Mann–Whitney test; P-value is indicated in the figure.

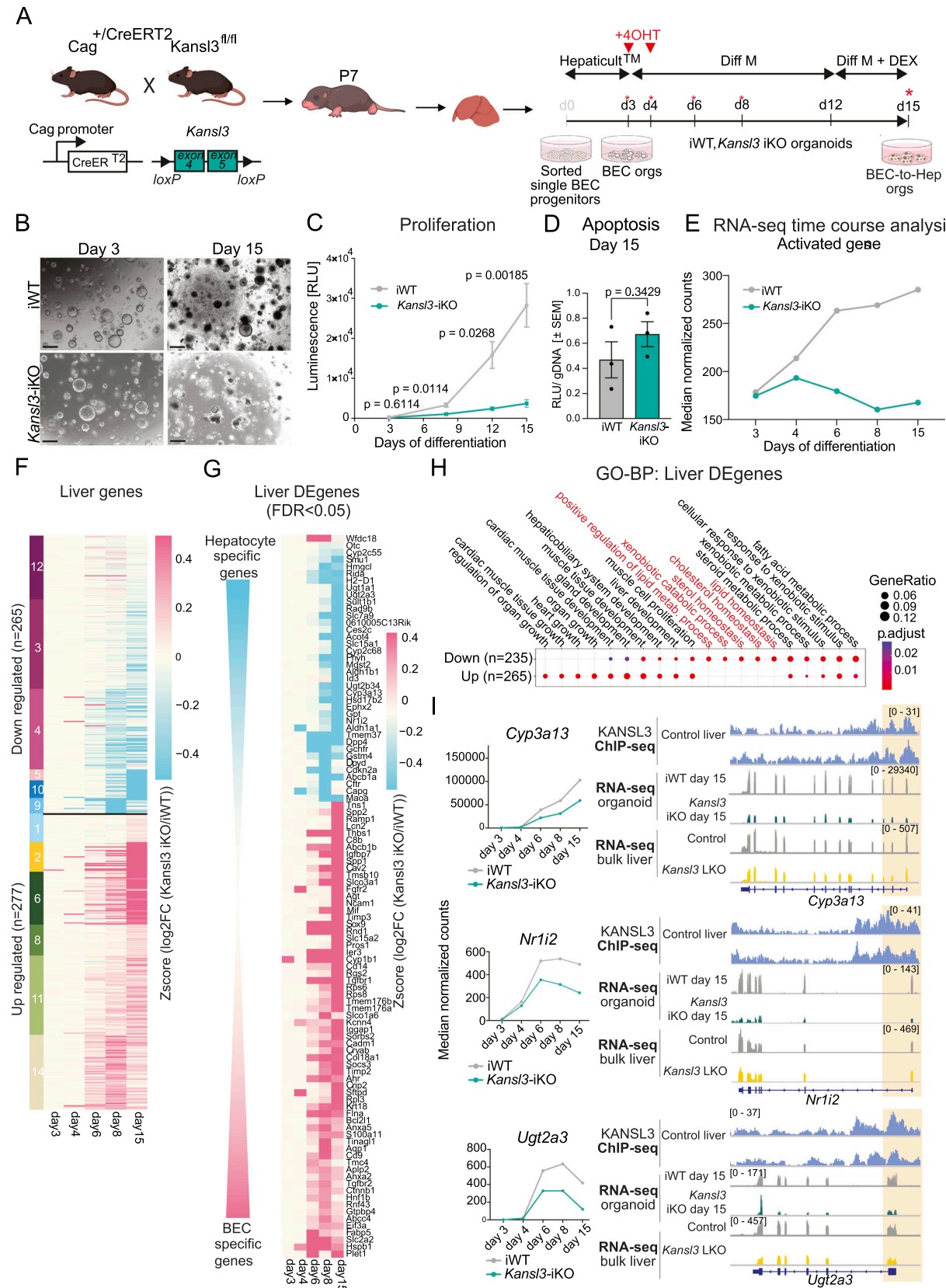

**Figure 5. KANSL3 determines hepatocyte differentiation of ductular organoids.**
**(A)** Schematic showing the generation of ductal organoids from neonatal and fetal liver organoids from E12.5 Kansl3^fl/fl (iWT) and Kansl3^fl/fl Cag-Cre-ERT2^T/+ (Kansl3-iKO) neonatal mice allowing for Cre-mediated recombination by 4-hydroxytamoxifen treatment (4OHT) (red arrows). Ductal organoid formation was performed from

### KANSL3 is important hepatocyte differentiation of fetal liver organoids

Given the early manifestation of the mouse phenotype, we hypothesize that KANSL3 may play a role in hepatocyte differentiation/maturation during fetal liver development. Although no marked differences were detected at the tissue level in E17.5 *Kansl3* LKO embryos (Figs 3A and B and S5B), molecular changes may still be present. To investigate the role of KANSL3 in hepatocyte differentiation from hepatoblasts, we generated fetal liver (FL) organoids from *Kansl3*^fl/fl^ and *Kansl3*^fl/fl^ and *Cag-CreERT2*^T/+^;*Kansl3*^fl/fl^ E12.5 embryos following an established protocol (Prior et al, 2019). Unsorted fetal liver cells were cultured for 14 d to differentiate into hepatocyte-like (Hep-like) FL organoids (Fig 6A). iWT Hep-like FL organoids showed densely budding structures characteristic of embryonic hepatocyte organoids (Prior et al, 2019) (Fig 6B). KANSL3 knockout was induced on day 3 of differentiation, leading to observable growth defects in organoids on day 14 (Fig 6B). In contrast to BEC-to-Hep organoids, the expression of hepatocyte markers *Alb*, *Hnf4a*, and *Cyp3a11* remained unchanged (Fig 6C). Nevertheless, mitochondrial genes such as *Mrpl11* and *Ndufb3* were significantly down-regulated, which is consistent with gene expression changes across both hepatic in vivo and in vitro models (Figs 6C and S3D). Notably, *Kansl3* iKO organoids exhibited reduced CYP3A4 activity (Fig 6D) and albumin secretion (Fig 6E), revealing impaired hepatocyte function.

Taken together, we demonstrated that KANSL3 loss disrupts the normal differentiation process of hepatocytes in both Hep-like FL and BEC-to-Hep organoids. Our results emphasize KANSL3's crucial role in hepatocyte differentiation and function, both during BEC transdifferentiation and during hepatoblast differentiation into hepatocytes, highlighting roles of KANSL3 during liver development and disease.

## Discussion

Our study reveals KANSL3 as a key regulator of hepatocyte function and cellular differentiation. Hepatocyte-specific deletion of KANSL3 in neonatal mice leads to rapidly progressing liver disease, characterized by biliary hyperplasia and fibrosis, indicating defects earlier in development. We identified KANSL3 as an important transcriptional regulator of core hepatocyte functions—including lipid and steroid metabolism and xenobiotic processing—and demonstrated that it controls its target genes via histone acetylation (Fig 6F).

We propose that the observed mouse phenotype arises from a combination of impaired hepatocyte differentiation during development and defective BEC transdifferentiation into hepatocytes later in life. Impaired fetal hepatocyte differentiation produces dysfunctional hepatocytes with reduced fitness and increased apoptosis, leading to a diminished hepatocyte population. This loss triggers a ductular reaction, and although scRNA-seq data indicate BEC-to-hepatocyte transition occurs in neonatal animals, BEC maturation into functional hepatocytes is disrupted, as confirmed by in vitro organoid differentiation assays. Based on our current data, we can only speculate whether KANSL3 plays a role in BEC transdifferentiation during adult liver regeneration. Because our analyses focused on neonatal animals - representing a developmental stage of liver maturation - it will be critical to investigate the effects ofKANSL3 depletion on BEC-to-hepatocyte transitions in adult liver disease models characterized by acute or chronic hepatocyte injury and impaired hepatocyte-mediated regeneration. Such studies will be essential to fully understand KANSL3's role in hepatic specification and liver repair.

Our gene expression data show that KANSL3 depletion affects different gene expression networks in either transdifferentiation or hepatocyte development in vitro. However, our results suggest that KANSL3 plays a crucial role in regulating gene expression in both processes. Hepatocyte differentiation in the fetal liver is regulated by a gene regulatory network of core liver transcription factors, including *HNF-1*, CCAAT/enhancer binding protein (C/EBP), *HNF-3*, *HNF-4*, and *HNF-6* (Duncan et al, 1998; Kyrmizi et al, 2006). In *Kansl3* LKO mice, hepatocytes are considerably depleted but not completely lost, suggesting that the gene regulatory network is compromised, but not completely disrupted. Given that the NSL complex

---

single BEC progenitor cells isolated by FACS sorting (see Fig S7B for sorting scheme) through a 3D culture in HepatiCult. BEC-to-Hep (hepatocyte) differentiation was induced by the addition of hepatocyte differentiation media (Diff M) over the course of 12 d. For the last 3 d, the media were supplemented with dexamethasone. Downstream experiments were performed at indicated time points. **(B)** Brightfield images of iWT and *Kansl3*-iKO BEC-to-Hep organoids at day 3 and day 15 of hepatocyte differentiation. Images were taken at 10x magnification; scale bar: 200 μm. **(C)** 3D RealTime-Glo MT cell viability assay showing proliferation (as relative luminescence units [RLUs]) of iWT (n = 4) and *Kansl3*-iKO (n = 4) BEC-to-Hep organoids during hepatocyte differentiation. Significance was determined using two-way ANOVA with the Geisser–Greenhouse correction and Tukey's multiple comparisons test. Exact *P*-values are indicated in the figure. **(D)** Caspase-Glo 3/7 3D assay showing apoptosis of iWT (n = 4) and *Kansl3*-iKO (n = 4) organoids at day 15 of hepatocyte differentiation. Blank corrected RLUs were normalized by the total gDNA amount. Significance was determined by the Mann–Whitney test. The exact *P*-value is indicated in the figure. **(E)** Mean normalized expression of all down-regulated genes at any time point in *Kansl3*-iKO BEC-to-Hep organoids (FDR:0.05, n = 666; Fig S8A, clusters 1,3,5,6,7), as identified by time-course RNA-seq analysis on days 3, 4, 6, 8, and 15 of in vitro hepatocyte differentiation. A complete list of deregulated genes detected in at least one time point of hepatocyte differentiation in *Kansl3*-iKO BEC-to-Hep organoids is available in Table S5. **(F)** Clustering of the expression of mouse liver genes (n = 755, collated from Planas-Paz et al [2019], Ardisasmita et al [2022], and Pu et al [2023a]) (Table S6) in organoids at days 3, 4, 6, 8, and 15 of differentiation. A total of n = 265 genes were down-regulated (log₂FC < 0, no FDR cutoff) or n = 277 were up-regulated (log₂FC > 0, no FDR cutoff). Genes with unchanged expression (n = 213) are not included in the figure. Z-score depicts the scaled log₂FC expression of *Kansl3*-iKO over iWT organoids. **(G)** Log₂FC expression changes of liver genes detected in *Kansl3*-iKO organoids during hepatocyte differentiation compared with iWT controls differentially expressed in at least one time point (FDR < 0.05). The data are ordered according to log₂FC at day 15. Z-score depicts the scaled log₂FC expression. Up-regulated genes are enriched for BEC markers (red), whereas the down-regulated genes are enriched for hepatocyte markers (blue). **(H)** Gene ontology analysis of biological processes for mouse liver genes deregulated in *Kansl3*-iKO organoids during hepatocyte differentiation. Color code indicates the adjusted *P*-value results, and the size of dots indicates the gene ratio per GO term. GO terms specific to down-regulated genes upon KANSL3 depletion are highlighted in red. **(I)** Left panels: mean normalized expression of hepatocyte-specific genes *Cyp3a13*, *Nr1i2*, and *Ugt2a3* in iWT control and *Kansl3*-iKO organoids on days 3, 4, 6, 8, and 15 of in vitro hepatocyte differentiation. Right panel: representative tracks of KANSL3 ChIP-seq in control liver tissue (input subtracted signal), RNA-seq in iWT and *Kansl3*-iKO day 3 and 15 BEC-to-Hep organoids, and RNA-seq in control and bulk liver tissue of 3-wk-old mice for the genes *Cyp3a13*, *Nr1i2*, and *Ugt2a3*. An orange box highlights the promoter region.

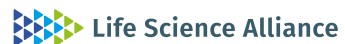

A

Cag$^{+/CreERT2}$ X Kansl3$^{fl/fl}$

E12.5

Cag promoter

CreER$^{T2}$

Kansl3

loxP — exon 4 — exon 5 — loxP

+4OHT

Hep-like M | Hep-like M -FGF7

d0 — d3 — d14 *

iWT, Kansl3-iKO organoids

Unsorted fetal liver cells

Hep-like FL orgs

B

**Day 3** | **Day 14**

Rep 1 | Rep 2 | Rep 3

iWT

Kansl3-iKO

C

**FL Hep-like organoids - Day 14 qPCRs**

mRNA relative to Hsp8 [± SEM]

*Kansl3* — p = 0.0286

*Mrpl11* — p = 0.0286

*Ndufb3* — p = 0.0286

*Alb* — p > 0.9999

*Hnf4a* — p = 0.7302

*Cyp3a11* — p = 0.3429

iWT | Kansl3-iKO

D

**CYP3A4 activity**

CYP3A4 activity [RLU/ug protein ± SEM]

p < 0.0001

iWT | Kansl3-iKO

E

**Albumin secretion**

Secreted Albumin/ ug protein [±SEM]

p = 0.0240

iWT | Kansl3-iKO

F

**Control**

Normal hepatocyte function

K3 / MOF / K2 / K1 — Metabolism Differentiation

Ac — ON

**Kansl3LKO**

Fibrosis ↑

Hepatocyte dysfunction

K3 / MOF / K2 / K1 — Metabolism Differentiation

Ac — OFF

Biliary hyperplasia ↑

is known to bind both promoters and enhancers (Chelmicki et al, 2014; Sun et al, 2023), it is likely that the NSL complex is required to activate target genes downstream of these core transcription factors. In addition, shared alterations in general cellular pathways—such as mitochondrial function, proliferation, and cellular motility—observed in both in vivo and in vitro models suggest these broader processes may also contribute to the phenotype.

Other NSL complex members have also been implicated in liver function with varying effects on liver health. For example, inducible hepatocyte-specific knockout of *Kat8* in adult mice showed no effect on hepatocyte function (Lei et al, 2021), whereas deletion of *Mcrs1* in hepatocytes results in severe liver cirrhosis and lethality within 3 wk post-induction because of accumulation of bile acids in the hepatic sinusoids (Garrido et al, 2022). Hence, individual NSL complex components seem to play distinct roles in regulating hepatocyte function and fitness. Because deletion of KANSL3 results in down-regulation of most NSL complex members, the *Kansl3* LKO phenotype may reflect a broader NSL complex deficiency. To parse KANSL3's specific functions during liver development versus adult homeostasis, inducible hepatocyte-specific (or BEC-specific) knockout models in adult animals will be necessary.

Interestingly, *Alb*Cre-driven knockout of *Ogt*, another NSL complex member, produces a phenotype similar to *Kansl3* LKO mice. Although developmental defects were ruled out in *Ogt* LKO mice, these animals develop liver disease and fibrosis by 4 wk of age (Zhang et al, 2019; Ortega-Prieto et al, 2024) with gene expression changes resembling those in 3-wk-old *Kansl3* LKO mice. Notably, OGT-mediated O-GlcNAcylation stabilizes KANSL3, sustaining NSL complex–mediated histone acetylation activity in vitro (Wu et al, 2017; Zhao et al, 2019), suggesting that the hepatic phenotypes observed in *Kansl3* and *Ogt* knockout animals may be mechanistically linked.

Collectively, we demonstrated a crucial role of the epigenetic regulator KANSL3 in hepatocyte differentiation in liver development and disease, identifying it as a potential target for therapeutic intervention in the treatment of liver disease.

# Materials and Methods

## Animals

All of the mice were kept in the animal facility of the Max Planck Institute of Immunobiology and Epigenetics. The mice were maintained under specific-pathogen-free conditions, with two to five mice housed in individually ventilated cages (Techniplast). The cages were equipped with bedding material, nesting material, a paper house, and a tunnel for refinement. The housing environment was carefully regulated, with a light cycle consisting of 14 h of light followed by 10 h of darkness. The ambient temperature was maintained at 22°C ± 2°C, and the humidity was kept at 55% ± 10%. Mice were handled using a tunnel to minimize stress and ensure their well-being. Mice were euthanized according to §4 (3) of the German Animal Protection Act, and all of the animal experiments were performed in accordance with the relevant guidelines and regulations, approved by the Regierungspräsidium Freiburg, Germany. This project was performed according to an "anzeigepflichtiges Versuchsvorhaben" (notifiable experimental project) and approved by the welfare officer S. Kunz, Max Planck Institute of Immunobiology and Epigenetics (licences Akh-iTo 2, Akh-iTo 3, Akh-iTo-Leber 1 and Akh-iTo-Leber 2). The *Kansl3*-floxed (Karoutas et al, 2019) mice have been previously described. The *Alb*Cre, *Cag-Cre-ERT2*, and mTmG reporter mice were purchased from The Jackson Laboratory. Genotyping was undertaken through standard PCR using the primers listed in Table S7.

## Histology and MASH activity scoring (MAS)

Livers from neonatal (P7), 3-wk-old (P21), and 90-wk-old *Kansl3* LKO, *Mof* LKO, and *Kansl3*^fl/fl control mice were dissected and fixed in 4% PFA in PBS, dehydrated, and embedded in paraffin. *Kansl3* LKO, *Mof* LKO, and *Kansl3*^fl/fl control E17.5 embryos were collected and killed by cervical dislocation, fixed in 4% PFA in PBS, dehydrated, and embedded in paraffin. Sections of 6-μm thickness were cut and stained with hematoxylin and eosin (VWR) and picrosirius red (Sigma-Aldrich). The images were captured with an Axio Imager Apotome microscope using a 20x objective and an AxioCam MRc color charge-coupled device (CCD) camera (Carl Zeiss). Images were analyzed with the AxioVision software (Carl Zeiss) for Axio Imager Apotome. The area of picrosirius red staining was quantified using the ImageJ macro for quantifying stained liver tissue with standard parameters (https://imagej.net/ij/docs/examples/stained-sections/index.html); for the analysis, ImageJ2 version 2.14.0/1.54f was used. MA (MASH/MAFLD activity) scoring using an adapted scoring system for rodents (Liang et al, 2014) was performed by AnaPath Services GmbH.

**Figure 6.  Loss of KANSL3 affects hepatic differentiation of fetal liver organoids.**
**(A)** Schematic showing the generation of ductal organoids from neonatal and fetal liver organoids from E12.5 *Kansl3*^fl/fl (iWT) and *Kansl3*^fl/fl *Cag-Cre-ERT2*^T/+ (*Kansl3*-iKO) mice allowing for Cre-mediated recombination by 4-hydroxytamoxifen treatment (4OHT) (red arrows). Fetal liver (FL) organoids were generated from unsorted fetal liver cells. Hepatocyte differentiation media (Hep-like M) were added for 14 d including the removal of FGF7 on day 3, resulting in FL Hep-like organoids. Downstream experiments were performed at indicated time points. **(B)** Brightfield images of one replicate of iWT and *Kansl3*-iKO FL Hep-like organoids at day 3 and three replicates at day 15 of Hep-like differentiation. Images were taken at 10× magnification, scale bar: 200 μm. **(C)** qRT–PCR of iWT and *Kansl3* iKO fetal liver organoids at day 14 of Hep-like differentiation of *Kansl3*, *Ndufb3*, *Mrpl11*, *Alb*, *Hnf4a*, and *Cyp3a11* mRNA expression. Expression levels are normalized to *Hsp8* expression. Significance was scored using the Mann–Whitney test, and the exact *P*-values are indicated in the panels. **(D)** P450-Glo CYP3A4 activity assay of day 14 iWT (n = 4) control and *Kansl3*-iKO (n = 5) FL Hep-like organoids. RLU was normalized to total protein. Significance was determined by the Mann–Whitney test, and the *P*-value is indicated in the figure. **(E)** Albumin secretion assay of day 14 iWT (n = 4) control and *Kansl3*-iKO (n = 4) FL Hep-like organoids. The amount of secreted albumin into the media was normalized to total protein. Significance was determined by the Mann–Whitney test, and the *P*-value is indicated in the figure. **(F)** Working model showing the importance of KANSL3 in hepatocytes. Deleting KANSL3 in hepatocytes driven by *Alb*Cre leads to early-onset liver disease, characterized by biliary hyperplasia (ductular reaction) and fibrosis. Loss of KANSL3 interferes with hepatocyte differentiation in vivo. Mechanistically, KANSL3 binds to the promoter of genes important for hepatocyte function and cellular differentiation. We conclude that epigenetic regulation by KANSL3 has a direct impact on gene expression in hepatocytes, which is essential for maintaining hepatocyte homeostasis.

### Immunofluorescence of cryotissue sections

Livers from 3-wk-old (P21) *Kansl3* LKO, *Mof* LKO, and *Alb*Cre control mice were dissected and embedded in OCT embedding matrix and frozen on dry ice. Sections of 6-$\mu$m thickness were cut, air-dried, and fixed in 1% PFA for 20 min. Three brief washes in 1x PBS were followed by permeabilization in 0.2% Triton in 1x PBS for 20 min and briefly washed again three times in 1x PBS. Sections were blocked in 0.25% gelatin solution for 1 hr at RT and incubated with primary antibodies overnight at 4°C in a humidity chamber. The next day, sections were washed three times in 1x PBS for 10 min and the secondary antibody was added for 1 hr at RT. After 2 × 10 min washes with 1x PBS, nuclei were stained with DAPI (1:2,000) for 20 min at RT. Finally, the slides were washed with PBS wash followed by deionized water and mounted using Vectashield (VWR). The images were captured on an LSM 900 microscope equipped with an Airyscan 2 detector (Carl Zeiss) using a (NA: 0.8) objective and analyzed with Zen Blue software (Carl Zeiss). The area of Krt19 staining was quantified using the ImageJ macro for quantifying stained liver tissue with standard parameters (https://imagej.net/ij/docs/examples/stained-sections/index.html); for the analysis, ImageJ2 version 2.14.0/1.54f was used.

### Oil Red O staining of cryotissue sections

Livers from 3-wk-old (P21) *Kansl3* LKO and *Alb*Cre control mice were dissected and embedded in OCT embedding matrix and frozen on dry ice. Sections of 6-$\mu$m thickness were cut, air-dried, and fixed in 1% PFA for 1 min. Sections were washed under running tap water for 10 min, rinsed in 60% isopropanol for 1 min, and stained in Oil Red O staining solution (six parts of 0.6% Oil Red O dye in isopropyl alcohol and four parts of water) for 15 min. After an additional rinse in 1% isopropanol, nuclei were lightly stained with hematoxylin. The images were captured with an Axio Imager Apotome microscope using a 20x objective and the AxioCam MRc color charge-coupled device (CCD) camera (Carl Zeiss). Images were analyzed with AxioVision software (Carl Zeiss) for Axio Imager Apotome.

### Generation and culture of the HA-dTAG-KANSL3 cell line

HA-dTAG-KANSL3 cells were generated according to a published protocol (Damhofer et al, 2021; Radzisheuskaya et al, 2021) with modifications. In brief, the guide RNA was designed using the Alt-R CRISPR HDR Design Tool from IDT, and also, the single-stranded DNA (ssDNA) donor fragment, which comprises a 181-bp left homology arm, 2x HA tags, GSG, FKBP$^{F36V}$, GSG, 181-bp right homology arm, was ordered from IDT. The guide RNA (combining Alt-R crRNA [100 $\mu$M] and fluorescently labeled Alt-R tracrRNA with ATTO 550 [100 $\mu$M] in 1:1 ratio)-Alt-R Cas9 enzyme (61 $\mu$M) was assembled into a ribonucleoprotein (RNP) complex in vitro in accordance with the manufacturer's protocol (IDT). The donor fragment (0.25 $\mu$g) and the RNP complex within the P3 nucleofection buffer (Lonza) were nucleofected into the 50,000 Hepa1–6 cells using the 4D-Nucleofector system (Lonza) with the EH-100 program in P3 buffer. The nucleofected positive cells (i.e., ATTO 550+) were sorted into the individual wells using FACS sorting 48 hrs post-nucleofection. Each propagated single-cell clone was then screened for the target insert region by PCR. The positive clones were subsequently validated for the presence of the correct insert by Sanger sequencing (with reverse primer). Primer sequences are listed in Table S7.

### Lentivirus-mediated ectopic expression

Full-length FLAG-tagged human KANSL3 was cloned into the pCHDblast MCSNard plasmid (Addgene vector no. 22661) using the Xba I and Not I cloning sites (NEB). The "empty vector" contained a 22–base pair (bp) fragment inserted into the same vector but lacked an open reading frame. This vector contains a cytomegalovirus (CMV) promoter and a blasticidin selection marker. Lentivirus was produced as previously described, and Hepa1-6 cells were infected 1 d after seeding (Chelmicki et al, 2014). Selection began 24 hrs after infection and was carried out over 3 d using blasticidin-HCl (2 $\mu$g/ml) (A11139-03; Gibco). On the 2nd d of selection, cells were treated either with dTAG-NEG or with dTAG13, and 24 hrs later, cells were harvested.

### Generation and culture of ductal organoids

Single-cell suspensions for the isolation of BECs from neonatal livers of *Kansl3*$^{fl/fl}$ and *Cag-CreERT2*$^{T/+}$;*Kansl3*$^{fl/fl}$ mice were prepared as described in the Methods section Single-cell RNA-seq. For the putative BEC progenitor cells, Zombie NIR (423105; BioLegend)–negative (live cells), CD45/CD31$^-$, and EpCAM$^{high}$/Tspan8$^{high}$/CD24$^+$ cells were sorted into cold basal media (advanced DMEM/F-12 (12634028; Gibco), 1x GlutaMax (35050038; Gibco), 10 mM Hepes (15630056; Gibco), 1x Pen/Strep (1514012; Gibco)) supplemented with 10 $\mu$M Rock inhibitor (Y27623; Sigma-Aldrich) on a BD FAC-Symphony S6 flow cytometer. In addition, CD45/CD31$^-$/EpCAM$^{high}$/Tspan8$^{medium}$ and CD45/CD31$^-$/EpCAM$^{high}$/Tspan8$^{low}$ cell populations were isolated as controls. FACS analyses were conducted by FlowJo v10.10.0. Cells were centrifuged at 550$g$ for 5 min at 8°C. The resulting pellet was resuspended in an appropriate volume of Matrigel (356231; Corning). For organoid formation, 1,000 cells were seeded per 50 $\mu$l of Matrigel. A 50 $\mu$l drop of the cell–Matrigel mixture was added to each well of a 24-well plate (CLS3738; Corning). The plate was then placed in a 37°C incubator for 15 min to allow the Matrigel to solidify. After solidification, 500 $\mu$l of HepatiCult media (06030; StemCell Technology) supplemented with 10 $\mu$M Rock inhibitor was added on top. The plate was incubated at 37°C in a 5% $CO_2$ incubator for 3 d. After the incubation period, the media were replaced with fresh HepatiCult. The cultures were split into new Matrigel at a 1:4 ratio every 3 d. To dissolve the Matrigel domes, 10 ml of ice-cold basal media was added, and the mixture was pipetted with a P1000 pipette to disaggregate the organoids. The suspension was then centrifuged at 400$g$ for 5 min at 8°C. This wash step was repeated twice to ensure the complete removal of residual Matrigel. The organoids were subsequently resuspended in fresh Matrigel, and domes were covered with HepatiCult. Organoids were cryopreserved in Recovery Cell Culture Freezing Medium (12648010; Gibco). See Table S7 for antibody information.

## Hepatocyte differentiation of ductal organoids (BEC-to-Hep)

To set up hepatocyte differentiation experiments, ductal organoid cultures (described above) were used. To dissolve the Matrigel domes, 10 ml of ice-cold basal media was added, and the mixture was pipetted with a P1000 pipette to disaggregate the organoids. The suspension was then centrifuged at 400*g* for 5 min at 8°C. This wash step was repeated twice to ensure the complete removal of residual Matrigel. The cells were then resuspended in 2 ml of prewarmed 1x TrypLE (12604013; Gibco), and the suspension was pipetted up and down using a P1000 pipette to facilitate cell dissociation. After this, the cell suspension was incubated in a 37°C water bath for 15 min. During the incubation, the cells were gently pipetted up and down every 5 min to ensure complete dissociation. After the incubation period, 8 ml of basal medium was added to the cell suspension to neutralize the TrypLE. The cell mixture was then centrifuged at 600*g* for 3 min at 8°C. The resulting pellet was resuspended in 500 μl of basal medium. To remove any cell clumps and ensure a single-cell suspension, the cell solution was filtered through blue cap FACS tubes (CLS352235; Corning) and briefly spun down. The suspension was then pipetted up and down to mix thoroughly before counting the cells. Cells were plated at a density of 4,000 cells per 50 μl of Matrigel. Once the Matrigel was solidified, 500 μl of HepatiCult media supplemented with 10 μM Rock inhibitor was added. After 3 d, the media were replaced with differentiation media, which were freshly prepared following a published protocol (Huch et al, 2015). The media were changed daily for 12 d. According to the experimental setup, the differentiation media were supplemented with 2 μM 4OHT (SML1666; Sigma-Aldrich) at days 3 and 4.

## Generation of fetal liver Hep-like organoids

Fetal liver Hep-like organoids were generated according to a published protocol (Prior et al, 2019). In brief, fetal livers of E12.5 embryos of *Kansl3*^fl/fl and *Cag-CreERT2*^T/+;*Kansl3*^fl/fl mice were dissected and digested (0.125 mg/ml collagenase clostridium histolyticum [Sigma-Aldrich], 0.125 mg/ml Dispase [Sigma-Aldrich], and 0.1 mg/ml DNase I [Roche] in wash buffer) (see the ductal organoid protocol) for 65 min in a 37°C water bath. Single cells were passed through a 40-μM cell strainer, pelleted, and resuspended in basal media (see the ductal organoid protocol). 5,000 cells were seeded per 50 μl of Matrigel (356231; Corning) and cultured in Hep-like media (advanced DMEM F12 supplemented with penicillin/streptomycin, GlutaMax, and Hepes [Gibco], 1x B27 [Gibco], 1.25 mM n-acetylcysteine [Sigma-Aldrich], 10 mM nicotinamide [Sigma-Aldrich], 100 ng/ml FGF10 [PeproTech], 100 ng/ml FGF7 [PeproTech], 50 ng/ml HGF [PeproTech], 10 nM gastrin [PeproTech], 50 ng/ml EGF [PeproTech], 1 nM A83-01 [Tocris Bioscience], 3 μM CHIR 99021 [Tocris Bioscience], 15% 3dGRO R-Spondin-1 Conditioned Media [Sigma-Aldrich], and 10 μM Rock inhibitor Y-27632 [Sigma-Aldrich]). From day 3 onward, the culture medium FGF7 was removed from the media and supplemented 2 μM 4OHT was added to induce *Kansl3* knockout. Hep-like organoids were analyzed on day 14 of differentiation.

## 3D RealTime-Glo MT cell viability assay

Hepatocyte differentiation was performed with minor modifications to the previously described method. Specifically, 1,000 single iWT (n = 4) or *Kansl3*-iKO (n = 4) ductal cells were seeded into 10-μl Matrigel domes within white, clear-bottom 96-well plates (CLS3610; Corning) and covered with 80 μl HepatiCult. Each cell line was seeded in triplicate wells as technical replicates. Organoid proliferation during hepatocyte differentiation was assessed using 3D RealTime-Glo MT Cell Viability Assay (G9711; Promega). This assay involved adding 1x MT Cell Viability Substrate and 1x NanoLuc Enzyme to the culture media, followed by a 2-hr incubation period. Luminescence was then measured using Victor Nivo Plate Reader. After recording, the media were replaced with fresh media. Measurements were taken on days 3, 4, 6, 8, and 15. For analysis, values were blank-subtracted, and the average of the three technical replicates per cell line was used for further calculations.

## Caspase-Glo 3/7 3D assay

Hepatocyte differentiation was carried out with slight modifications to the previously described protocol. A total of 4,000 single iWT (n = 3) or *Kansl3*-iKO (n = 3) ductal cells were seeded into 50-μl Matrigel domes in 24-well plates (CLS3738; Corning) and overlaid with 500 μl HepatiCult. Each cell line was seeded in triplicate wells as technical replicates. Apoptosis was assessed on day 15 of differentiation using the Caspase-Glo 3/7 3D assay (G8981; Promega) following the manufacturer's instructions. After luminescence readings were taken, genomic DNA was extracted from each well using DNeasy Blood and Tissue Kit (69504; QIAGEN), according to the manufacturer's protocol. Blank-subtracted luminescence values were normalized to the total gDNA, and the average from the three technical replicates per cell line was used for further calculations.

## P450-Glo CYP3A4 activity assay

Fetal liver (FL) Hep-like organoid differentiation was performed as described above. Each cell line was seeded in duplicates. CYP3A4 enzymatic activity was assessed on day 14 of differentiation using the P450-Glo CYP3A4 activity assay with luciferin-IPA (V9001; Promega) following the manufacturer's instructions. After luminescence readings were taken, the organoids in each well were lysed in 1x RIPA buffer and total protein was measured using Pierce Dilution-Free Rapid Gold BCA Protein Assay according to the manufacturer's protocol. Blank-subtracted luminescence values were normalized to the total protein, and the average from the two technical replicates per cell line was used for further calculations.

## Albumin secretion assay

FL Hep-like organoid differentiation was performed as described above. Each cell line was seeded in duplicates. The media on Hep-like organoids were changed on day 11, and the secreted albumin in the media was measured using BCG Albumin assay Kit (MAK124;

Sigma-Aldrich) in a 200 µl format according to the manufacturer's instructions. The organoids in each well were lysed in 1x RIPA buffer, and total protein was measured using Pierce Dilution-Free Rapid Gold BCA Protein Assay according to the manufacturer's protocol. Blank-subtracted luminescence values were normalized to the total protein, and the average from the two technical replicates per cell line was used for further calculations.

**Quantitative RT–PCR**

For qRT–PCR analyses, RNA was extracted from bulk liver tissue or Hepa1 –6 cells using Quick-RNA Microprep Kit (R1051; Zymo). For organoids, the RNA was extracted using Single Cell RNA Isolation Kit (Norgen Biotek) or miRNeasy Tissue/Cells Advanced Micro Kit (QIAGEN). Reverse transcription reactions were carried out using GoScript Reverse Transcriptase (A5001; Promega) as per the manufacturer's instructions. The transcript levels were quantified using SYBR Green chemistry on an LC480 instrument (Roche). The primer sequences are provided in Table S7.

**Generation of extracts from bulk liver tissue**

100 µl of the ice-cold extraction buffer, HMG150 buffer (25 mM Hepes [pH 7.6], 12.5 mM $MgCl_2$, 10% glycerol, 150 mM KCl, 0.5% Tween-20, protease inhibitor cOmplete mini [04693159001; Roche], and PhosSTOP [04906837001; Roche]), was added to 10 mg of frozen liver tissue. The tissue in the extraction buffer was then homogenized using a motorized pestle mixer for 20 sec at 700g. The suspension was incubated for 20 min at 4°C on a rotating wheel, followed by a short centrifugation for 1 min at 2,000g at 4°C. The whole lysate was transferred into a 96-well plate for mild sonication in a PIXUL sonicator (Active Motif) with the settings: N: 50, PRF:1, Burst: 20, Time: 2:30. After sonication, the samples were pooled and centrifuged at 14,000g for 15 min at 4°C. The supernatant containing the whole-cell extracts was collected into a new tube, and protein concentration was determined using a BCA assay (Pierce).

**Western blot**

Western blot analyses were performed using standard methods. Briefly, 30 µg of whole-cell extracts was run on a 4–12% gradient gel (Thermo Fisher Scientific) and proteins were transferred to a polyvinylidene difluoride (PVDF) membrane (10600021; Amersham). The membrane was blocked in 5% skim milk and incubated overnight with the appropriate primary antibody (Table S7). After washing, the membrane was incubated with the appropriate HRP-conjugated secondary antibody and developed using the Lumi-Light reagent (12015200001; Roche) with detection of chemiluminescence on a Bio-Rad ChemiDoc XRS+ instrument. Quantification and image analyses were carried out using ImageLab software (v5.2; Bio-Rad Laboratories).

**Serum analysis**

Blood from neonatal control, *Kansl3*, and *Mof* LKO mice was collected after decapitation from the neck wound. Blood from adult mice (3 wk old and 90 wk old) was obtained from the heart after cervical dislocation of the animals. Whole blood samples were incubated at RT for 20 min, and serum was isolated by 10-min full-speed centrifugation. The serum samples were analyzed using the following kits according to the manufacturer's instructions: Aspartate Aminotransferase (AST) Activity Assay Kit (MAK055; Sigma-Aldrich), Bilirubin Assay Kit (MAK126; Sigma-Aldrich), BCG Albumin Assay Kit (MAK124; Sigma-Aldrich), and High Sensitivity Triglyceride Fluorometric Assay Kit (MAK264; Sigma-Aldrich).

**Total cholesterol assay**

Total cholesterol levels in the livers of neonatal (P7) and 3-wk-old (P21) *Kansl3* LKO and *AlbCre* control mice were analyzed using Total Cholesterol Assay Kit (STA-390; Cell Biolabs) according to the manufacturer's instructions with minor modifications. Briefly, the frozen liver tissues were washed with 300 µl of cold PBS and homogenized using a motorized pestle mixer. 40 µl of the homogenized tissue was separated and mixed with RIPA buffer for later protein quantification. The remaining cells were spun briefly and suspended in 200 µl of chloroform:isopropanol:NP-40 (7:11:0.1). The extract was then centrifuged for 10 min at 15,000g, after which the organic phase was transferred to a new tube and air-dried ON under a hood. The dried lipids were dissolved in 200 µl of 1X Assay Diluent by sonication (20% duty cycle, 15 times) until the solution was homogeneous. The extracted samples were diluted 1:10 with the 1X Assay Diluent and subsequently quantified in duplicate, following the manufacturer's protocol.

**Quantification of hepatocytes and BECs in P7 mTmG reporter mice**

Livers from mTmG control (*Alb*Cre+) and *Kansl3* LKO mice were dissected from P7 neonatal mice, minced, and digested (0.25 mg/ml collagenase clostridium histolyticum [Sigma-Aldrich], 0.1 mg/ml DNase I [Roche] in wash buffer) (see the ductal organoid protocol) for 1 hr at 37°C on a wheel. Cells were pelleted and resuspended in 1x TrypLE (Gibco) in a 37°C water bath for 10 min. Cells were pelleted and incubated in ACK lysis buffer (Gibco) for 2 min at RT. Cells were filtered through a 100-µM cell strainer, pelleted, and washed in the wash buffer. Cells were stained with 1:200 APC-EpCAM and 1:500 Zombie NIR (423105; BioLegend) on ice, in the dark for 30 min. Cells were washed twice in the wash buffer and analyzed on BD LSRFortessa X-20 Cell Analyzer.

**Annexin V staining**

Livers from mTmG control (*Alb*Cre+) and *Kansl3* LKO mice dissected from P7 neonatal mice were treated as described above. After APC-EpCAM/Zombie NIR staining, cells were washed in a staining buffer (BioLegend). The pellets were then resuspended in 100 µl 1x Annexin V binding buffer supplemented with 5 µl Annexin V-BV421 (640923; BioLegend) and incubated at RT for 5 min. Subsequently, apoptotic cells were analyzed on BD LSRFortessa X-20 Cell Analyzer.

## Bulk RNA-seq

### Liver tissue

Liver tissue samples were obtained from 3-wk-old *Alb*Cre (n = 6) and *Kansl3* LKO (n = 6) mice. The tissue was lysed in 300 µl of TRIzol (15596018; Thermo Fisher Scientific) using an electric pestle. To ensure consistency and eliminate batch effect, RNA extractions were performed simultaneously for all samples using Quick-RNA Miniprep Kit (R1055; Zymo). RNA quality was examined using 5,200 Fragment Analyzer (Agilent). Libraries were prepared using TruSeq Stranded Total RNA Library Prep Gold (20020599; Illumina) and sequenced on the NovaSeq 6000 instrument (Illumina).

### BEC-to-Hep organoids

iWT (*Kansl3*$^{fl/fl}$) and *Kansl3*-iKO liver organoids were harvested at days 3, 4, 6, 8, and 15 of hepatocyte differentiation. Three iWT and four *Kansl3*-iKO organoid lines from individual animals were used for this experiment. 2 µM 4OHT (SML1666; Sigma-Aldrich) was added to the media of all samples at days 3 and 4 for a total of 48 hrs. Organoids were washed three times in 5 ml of ice-cold basal media to remove residual Matrigel. RNA was extracted using Single Cell RNA Isolation Kit (Norgen Biotek). RNA quality was examined using the 5,200 Fragment Analyzer (Agilent). For day 3–8 samples, libraries were prepared using the Stranded mRNA Prep, Ligation kit (20040534; Illumina). For day 15 samples, libraries were prepared using NEBNext Single Cell/Low Input RNA Library Prep Kit for Illumina (E6420; NEB). All samples were sequenced on the NovaSeq 6000 instrument (Illumina).

### HA-dTAG-KANSL3 Hepa1-6 cells

24-hr dTAG-NEG− and dTAG-13−treated HA-dTAG-KANSL3 Hepa1-6 cells were directly lysed on the plate in 300 µl of TRIzol (15596018; Thermo Fisher Scientific). The experiment was performed in triplicates. To ensure consistency and eliminate batch effect, RNA extractions were performed simultaneously for all samples using the Quick-RNA Miniprep Kit (R1055; Zymo). RNA quality was examined using the 5,200 Fragment Analyzer (Agilent). Libraries were prepared using the Stranded mRNA Prep, Ligation kit (20040534; Illumina) and sequenced on the NovaSeq 6000 instrument (Illumina).

### RNA-seq analysis

The data were processed using the snakePipes mRNA-seq pipeline (v.2.7.2) (Bhardwaj et al, 2019). Adapters and low-quality bases (<Q20) were removed using TrimGalore (v.0.6.5) (https://github.com/FelixKrueger/TrimGalore) with the parameters "-q 20 --trim-n". The trimmed reads were aligned using STAR (v.2.7.4) (Dobin et al, 2013) to the GRCm38 (mm10) genome. The aligned reads were counted using featureCounts (v.2.0.0) (Liao et al, 2014). Bigwig files were created using deepTools bamCoverage (v.3.3.2) (Ramírez et al, 2016). The gene-level counts obtained from featureCounts were then used for differential expression analysis using DESeq2 (v1.26.0). Genes were considered to be differentially expressed with a *Q*-value cutoff of 0.05. Heatmaps of *z*-score were generated using the *pheatmap* package in R. GO enrichment analysis of misregulated genes was performed using ClusterProfiler (v.3.17.4) (Yu et al, 2012).

## ChIP-seq

### Fixation of liver tissue and Hepa1-6 cells

For *Alb*Cre liver tissue, a small proportion was finely chopped. The tissues were transferred into a Dounce homogenizer (loose pestle), covered with 1 ml of 1% formaldehyde in DMEM (Gibco), and dissociated with 3–4 strokes, followed by 15-min incubation at RT. During incubation, the tissue suspension was filtered using a nylon 70-µM cell strainer (352350; Falcon) to remove larger debris. 125 mM glycine final was added to the sample, and the suspension was pelleted for 5 min at 500*g*. Cell pellets were washed twice in ice-cold 1x PBS and aliquoted. Fixed cell pellets were stored at −80°C until further usage. HA-dTAG-KANSL3 Hepa1-6 cells were fixed in 1% formaldehyde in culture medium and incubated at RT on a rocking plate for 15 min. The medium was removed, and cells were washed twice with ice-cold 1x PBS. Cells were scraped off the plate, collected in ice-cold 1x PBS, and centrifuged at 2,000*g* for 5 min at 4°C. After an additional 1× PBS wash, cell pellets were stored at −80°C until use.

### Transcription factor ChIP-seq or ChIP-qPCR

Transcription factor ChIP was performed according to the published RELACS ChIP-seq protocol (Arrigoni et al, 2018) omitting the nucleus barcoding procedure. In brief, nuclei were isolated by mild sonication using the NEXSON-based nucleus isolation protocol. Cell pellets were thawed on ice and resuspended in ice-cold 1 ml of lysis buffer (10 mM Tris−HCl, pH 8, 10 mM NaCl, 0.2% IGEPAL CA-630, 1x protease inhibitor cocktail). The cell suspension was then transferred into 1 ml milliTUBE (Covaris) and sonicated in the Covaris instrument (E220) for 30 sec at peak power 75 W, duty factor 2%, and 200 cycles/burst. Nuclei were pelleted at 1,000*g* at 20°C for 5 min. The supernatant was discarded, and nuclear pellets were carefully resuspended in 0.5% SDS and incubated at RT for 10 min, followed by quenching of SDS by addition of Triton X-100 at 1.1% final concentration. 1x CutSmart buffer and 100x protease inhibitor cocktail were added to the chromatin and digested using CvikI-1 (5 U per 100,000 cells, R0710S; New England Biolabs) at 20°C for 16 hrs, with shaking at 800 rpm on an Eppendorf Thermomixer C. After the chromatin digestion, the nuclei were pelleted for 5 min at 1,000*g* and washed in cold nucleus wash solution (10 mM Tris−HCl, pH 8, 0.25% Triton X-100, 0.2 mg ml$^{-1}$; BSA), and stored on ice. To confirm the enzymatic chromatin shearing efficiency, 5% of the nucleus solution was incubated together with a decrosslinking solution consisting of proteinase K, RNase A, and 5 M NaCl at 50°C for 30 min followed by incubation for 2 hrs at 65°C. DNA was purified using the MinElute PCR purification kit (QIAGEN) and analyzed on 5,200 Fragment Analyzer System (M5310AA) using HS NGS Fragment Kit (1–6,000 bp) reagents (Agilent). Digested nuclei were then lysed to perform transcription factor ChIP. A list of the antibodies is provided in Table S7. iDeal ChIP-seq Kit for Transcription Factors (C01010055; Diagenode) was used with modifications. In brief, 200 µl chromatin was incubated with the antibody for 10 h at 4°C, followed by a 3-hr incubation with protein G magnetic beads (Thermo Fisher Scientific). Wash steps were performed for 5 min per ChIP, after which DNA was recovered, deproteinized, and decrosslinked for 2 hrs at 65°C. DNA was purified using the MinElute PCR purification kit (QIAGEN). Either DNA was analyzed by qRT−PCR using SYBR Green chemistry on an LC480 instrument (Roche) (for primer sequences,

see Table S7) or DNA samples were USER-treated and PCR-amplified as described using components of the NEBNext Ultra II library preparation kit. At least two independent replicates were sequenced per condition.

### Histone modification ChIP-seq

Digested nuclei from Hepa1–6 cells were pelleted at 5,000*g* for 10 min and normalized to the concentration of 500,000 nuclei per 25 µl in 10 mM Tris–HCl, pH 8. Nucleus barcoding was performed as described in the RELACS protocol (Arrigoni et al, 2018). In brief, 1.5 µl of End Prep Enzyme Mix and 3.5 µl of reaction buffer, from the NEBNext Ultra II DNA library preparation kit (E7645L; NEB), were added to the nuclei, and then incubated at 20°C for 30 min followed by heat inactivation at 65°C for 5 min. Subsequently, 1.2 µl of hairpin adapters containing sample barcodes was added. Barcodes were ligated to the in situ digested chromatin by adding 15 µl of ligation master mix and 0.5 µl of ligation enhancer from the NEBNext Ultra II DNA library preparation kit (E7645L; NEB), followed by 15-min incubation at 30°C and 15-min incubation at 20°C. Each ligation reaction was inactivated by adding 300 mM NaCl final concentration. Barcoded nuclei from different samples were pooled together and pelleted at 5,000*g* for 10 min. Nuclei were resuspended in shearing buffer (10 mM Tris–HCl, pH 8, 0.1% SDS, 1 mM EDTA, 1x protease inhibitor cocktail) and lysed by soni-cation. The nucleus suspension was transferred into a Covaris microTUBE (520052) and sonicated for 5 min at peak power 105 W, duty factor 2%, and 200 cycles/burst. To remove debris, the chro-matin solution was centrifuged at 20,000*g* at 4°C for 10 min. Au-tomated ChIP was performed using the IP-Star platform (Diagenode) and the iDeal ChIP-Seq kit (C03010020; Diagenode) according to the manufacturer's instructions. In brief, 200 µl chromatin was incu-bated with the antibody for 10 hrs at 4°C, followed by incubation with protein A magnetic beads (Thermo Fisher Scientific). After washing, DNA was recovered, deproteinized, and decrosslinked for 2 hrs at 65°C. DNA was purified using the MinElute PCR purification kit (QIAGEN), USER-treated, and PCR-amplified as described using components of the NEBNext Ultra II library preparation kit. At least two independent replicates were sequenced per condition. For a list of the antibodies, see Table S7.

### ChIP-seq analysis

The data were processed using the snakePipes DNA-mapping and ChIP-seq pipelines (v.2.7.2). The DNA-mapping part was the same as for the ATAC-seq analysis (see above). For DNA mapping, adapters and low-quality bases (<Q20) were removed using TrimGalore (v.0.6.5; https://github.com/FelixKrueger/TrimGalore) with the parameters "-q 20 --trim-n". Reads were then mapped to the genome version GRCm38 with Bowtie2 (v.2.3.5) (Langmead & Salzberg, 2012). Reads that mapped to the blacklisted regions from the Encode Consortium (Amemiya et al, 2019) were discarded. Duplicated reads were also marked using Picard Mark-Duplicates (v.1.65; https://broadinstitute.github.io/picard/) and fil-tered out. In the end, only properly paired mapped reads and reads with a mapping quality over 3 were retained to generate BAM files. Peak calling was performed using MACS2 (v.2.2.6) (Zhang et al, 2008) with the *Q*-value to 0.001 using the input as a control. Bigwig files were created using deepTools bamCompare (v.3.5.0) (Ramírez et al,

2016) using the input normalization method and the log$_2$ ratio and subtract option.

### Single-cell RNA-seq

### Flow cytometry sorting

Neonatal mice were euthanized by decapitation, and livers were isolated. One piece of the liver was used for BEC isolation and the other for hepatocyte isolation. For BEC isolation, the liver piece was placed into an ice-cold wash buffer (DMEM, 1% FCS, 1x Pen/Strep [1514012; Gibco]). For BEC isolation, liver tissue was minced using a scalpel, washed twice in 10 ml ice-cold wash buffer, and then incubated in 10 ml of prewarmed digestion buffer (0.25 mg/ml collagenase [C2674; Sigma-Aldrich] in wash buffer) on a wheel at 37°C for 1 hr. Cells were spun down at 400*g* for 5 min and then incubated in 2 ml of 5x TrypLE solution (12604013; Gibco) for 10 min in a water bath. After adding 10 ml of wash buffer to quench the reaction, cells were centrifuged at 600*g* for 3 min. After centrifugation, the supernatant was removed, and 1 ml of ACK buffer (A1049201; Gibco) was added and the cells were in-cubated at RT for 2 min. 3 ml of the wash buffer was added, and the solution was filtered through a 70-µm filter (431751; Corning) into a fresh 15-ml Falcon tube. Samples were centrifuged again at 600*g* for 3 min, and the cell pellet was resuspended in 100 µl of antibody staining solution and incubated for 45 min. A list of antibodies is contained in Table S7. After the incubation, 1 ml of wash buffer was added to the cells, and the tubes were centrifuged at 600*g* for 2–3 min. This wash step was repeated once more. After the final wash, 500 µl of wash buffer was added, and the solution was transferred to blue-capped FACS filter tubes (352235; Corning) and the tubes were briefly centrifuged to collect the cells. Cells were sorted on a MoFlo flow cytometer into the ice-cold wash buffer. For the BEC population, Zombie NIR (423105; BioLegend)–negative (live cells), CD45/CD31/CD11b⁻, and EpCAM⁺ cells were sorted into the wash buffer. For the HSC enriched population, Zombie NIR (423105; BioLegend)–negative (live cells), CD45/CD31/CD11b/EpCAM⁻ cells were sorted into the wash buffer.

The other part of the liver was used for hepatocyte isolation. The liver tissue was placed into an ice-cold HBSS solution. The tissue was minced using a scalpel and washed twice with ice-cold HBSS with centrifugation at 120*g* for 5 min. The tissue was then incubated in 1 ml of prewarmed digestion buffer (0.8 mg/ml collagenase [C2674; Sigma-Aldrich] and 50 U/µl DNase I [AppliChem] in DMEM) on a wheel at 37°C for 30 min. The reaction was stopped by adding 5 ml of 2% BSA solution (in DMEM) and spun down at 200*g* for 5 min. After centrifugation, the pellet was resuspended in 1 ml of ACK buffer (A1049201; Gibco) and incubated at RT for 2 min. 5 ml of 2% BSA buffer was added, and the solution was centrifuged at 200*g* for 5 min. The pellet was then resuspended in 2% BSA buffer supple-mented with 1:500 of Zombie NIR (423105; BioLegend), filtered through blue cap FACS filters (352235; Corning), and sorted on a MoFlo flow cytometer into ice-cold 2% BSA buffer. For the hepatocyte enriched fraction, Zombie NIR–negative, large cells were sorted.

### RNA and library preparation from single cells

For the scRNA-seq experiment, BECs, hepatocytes, and HSC populations were isolated from three *Alb*Cre and three *Kansl3* LKO mice. For each population, the samples from each genotype were pooled. The BECs, hepatocytes, and HSC populations were mixed in a 1:1:1 ratio. This cell mix was directly subjected to 10x scRNA-seq library preparation. One 10x reaction was performed per genotype. A total of 16,000 cells per genotype were processed using the 10X Genomics protocol CG00052 RevB from Chromium Single Cell 3′ Library and Gel Bead Kit v2 (PN-120237). The libraries were quantified using a fragment analyzer and pooled for multiplexed sequencing on NovaSeq 6000.

### Single-cell RNA-seq analysis

The scRNA-seq data were processed using Cell Ranger (10X Genomics) (Zheng et al, 2017) to demultiplex and align the reads to the mm10 genome build, generating a raw read count matrix with barcodes representing individual cells and features corresponding to detected genes. The neonatal scRNA-seq data were processed using Cell Ranger v3.0.2. For both experiments, the counts from Cell Ranger were imported into R (version 4.1) and data were processed using the Seurat suite version 4.0. Low-quality cells were filtered out by keeping only cellular barcodes with greater than 500 UMIs per cell, greater than 500 genes per cell, and less than 5% of reads mapping to mitochondrial genes. Data were log-normalized and scaled, and the 2,000 most variable genes were identified. We reduced dimensionality using PCA to 50 to explain the majority of variation in the data. PC loadings were used as input for a graph-based approach to cluster cells by type and as input for Uniform Manifold Approximation and Projection.

### Fate-bias analysis and pseudotime inspection

Fate-bias probability was calculated using FateID (Herman & Sagar, 2018). The *FateID* algorithm applies an iterative random forest classification to quantify fate bias in naïve progenitors using cells classified in *RaceID3* and *StemID2* as training sets. The expression data and cluster partitions filtered by *RaceID3* were used as input data for the fate-bias analysis. The endpoints of the differentiation trajectories are determined by *RaceID3/Seurat* clusters, that is, hepatocyte and HSC endpoints (*tar* input). The *x* input was based on the expression data extracted from Seurat, and the *y* input reflected the Seurat clusters. The fate bias toward target clusters was calculated by *fatebias()* from the *FateID* algorithm. Dimensional reduction representations were computed on the basis of Uniform Manifold Approximation and Projection maps using *compdr()*. The results from *fatebias()* and *compdr()* were used to generate the principal curve using *prcurve()*. The fraction threshold of random forest votes required for the inclusion of a given cell in the principal curve was set to 0.3.

### Gene expression changes along the hepatocyte pseudotime

To eliminate lowly expressed genes on the trajectory, the RaceID-normalized transcript expression values were filtered with *filterset()*. The following parameters were applied: ≧2 normalized expression counts per gene (*minexpr* input), genes expressed at

least in one cell (*minnumber* input), and a vector containing the cell names extracted from the principal curve (*n*, input). The self-organizing map (SOM) of the pseudotemporal order was generated using *getsom()* with the following inputs: the filtered gene list, *n* = 1,000 maximum nodes (default value), and *α* = 0.5. The *getsom()* returns a data frame with smoothed and normalized expression profiles and *z*-score–transformed pseudotemporal expression profiles. The SOM was further processed by *prsom()*. The *prsom()* groups the nodes generated by *getsom()* into larger nodes, in which genes having higher than 0.85 correlation of the SOM *z*-scores are aggregated into the same node. The minimal number of genes per node was set to 5. The processed SOM was plotted with *plotheatmap()*.

### Quantification, statistical analysis, and data illustration

Data are presented as means ± SEM or as indicated in the figure legends. The exact replicate numbers are mentioned in the respective Materials and Methods sections or figure legends. All statistical analyses were performed using Prism 10.2.2 software (GraphPad) unless otherwise specified. The exact *P*-values are shown in the figures. Details of statistical tests are reported in the figure legends. Unless otherwise noted, figures were generated using Prism 10.2.2 (GraphPad) and Adobe Illustrator (2024). Schematic illustrations were created with BioRender, which uses an integrated AI-assisted design model.

## Data Availability

All data needed to evaluate the conclusions in this article are present in the article and/or the Supplementary Materials. The raw and processed data in this article were uploaded to GEO. The ChIP-seq can be accessed under the accession number GSE277828, the RNA-seq data under GSE277829, and the scRNA-seq data under GSE277833. All other data supporting the findings of this study are available from the corresponding author on reasonable request.

## Supplementary Information

## Acknowledgements

We express our gratitude to Yilong Zhou, Rudi Großschedl, and Eirini Trompouki for their critical review of the article. We also thank Meri Huch for her insightful discussions and further feedback on the article. We thank Andrea Tannapfel for her evaluation of the histological specimen from old animals. Furthermore, we thank Fabian Hässler for his help with mouse experiments. We appreciate Ward Deboutte's assistance with the GEO data upload. The support provided by the MPI-IE core facilities for FACS, deep sequencing, imaging, and mouse care has been invaluable to this project. We thank the whole Akhtar lab near and far for their input and help over the years. This study was supported by the Deutsche Forschungsgemeinschaft (DFG, German Research Foundation) under Germany's Excellence Strategy (CIBSS—EXC-2189—Project ID 390939984) (to A Akhtar), DFG SFB 1381 (Project

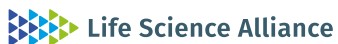

ID 403222702) (to A Akhtar), DFG CRC 992 (Project A02) (to A Akhtar), DFG CRC 1425 (Project P04) (to A Akhtar), and the Gottfried Wilhelm Leibniz Prize awarded to A Akhtar.

## Author Contributions

M Wiese: conceptualization, supervision, formal analysis, validation, investigation, visualization, methodology, project administration, and writing—original draft, review, and editing.

C Pessoa Rodrigues: data curation, software, formal analysis, validation, visualization, methodology, and writing—original draft, review, and editing.

ME Akbas: formal analysis, investigation, visualization, and methodology.

Y Sun: data curation, software, formal analysis, and visualization.

H Holz: investigation and resources.

JA Martinez Greene: investigation.

TH Tsang: formal analysis, investigation, and visualization.

C Bella: investigation and methodology.

K Ganter: investigation and resources.

T Stehle: investigation and resources.

M Shvedunova: writing—original draft.

A Akhtar: conceptualization, supervision, funding acquisition, and writing—original draft, review, and editing.

## Conflict of Interest Statement

The authors declare that they have no conflict of interest.

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
