## [Reviewer comments · Life Science Alliance]

KANSL3 directs transcriptional programmes essential for hepatic metabolism and differentiation

Meike Wiese, Cecilia Pessoa Rodrigues, Mehmet Akbas, Yidan Sun, Herbert Holz, Juan Martinez Greene, Tsz Hong Tsang, Chiara Bella, Kerstin Ganter, Thomas Stehle, Maria Shvedunova, and Asifa Akhtar

DOI: <https://doi.org/10.26508/lsa.202503238>

Corresponding author(s): Asifa Akhtar, Max Planck Institute of Immunobiology and Epigenetics

Review Timeline:

Submission Date:	2025-01-27
Editorial Decision:	2025-04-02
Revision Received:	2025-08-08
Editorial Decision:	2025-09-05
Revision Received:	2025-09-13
Accepted:	2025-09-16

Scientific Editor: Sarita Hebbar

Transaction Report:

April 1, 2025

Re: Life Science Alliance manuscript #LSA-2025-03238-T

Dr. Asifa Akhtar
Max Planck Institute of Immunobiology and Epigenetics
Max Planck Institute of Immunobiology and Epigenetics
Stübeweg 51
Freiburg 79108
Germany

Dear Dr. Akhtar,

Thank you for submitting your manuscript entitled "KANSL3 directs transcriptional programmes essential for hepatic metabolism and differentiation" to Life Science Alliance. The manuscript was assessed by three expert reviewers, whose comments are appended to this letter.

Overall, the reviewers concluded that the results are convincing to support a role of KANSL3 in liver development. However, they raised important concerns that need to be addressed for publication here. Some of these are listed here:

1. All reviewers noted that conclusions on the role of KANSL3 in adult liver maintenance/regeneration were not fully supported by the data shown. They suggested experiments to address this, or to modify the claims in the absence of new experimental data. Here we concur with the view of Reviewer 1 who felt the strength of the work lies in development/differentiation roles of KANSL3.
2. Reviewers 2 and 3 sought confirmation of NSL-mediated acetylation
3. Reviewers 1 and 3 suggested experiments and/or changes to the text to better describe the cellular and metabolic phenotypes in mice and organoids.
4. All the reviewers requested further elaboration on scRNA seq data including differences in numbers of up- versus down-regulated genes, shortlisted genes, and genes with metabolic functions.

Given the overall recommendations, we would like to invite you to submit a revised manuscript addressing the Reviewers' comments. Please include a letter addressing all the reviewers' comments point by point.

Thank you for this interesting contribution to Life Science Alliance. We are looking forward to receiving your revised manuscript.

Sincerely,

Sarita Hebbar, PhD
Scientific Editor
Life Science Alliance
<http://www.lsjournal.org>

- A letter addressing the reviewers' comments point by point.
- An editable version of the final text (.DOC or .DOCX) is needed for copyediting (no PDFs).
- High-resolution figure, supplementary figure and video files uploaded as individual files: See our detailed guidelines for preparing your production-ready images, <https://www.life-science-alliance.org/authors>
- Summary blurb (enter in submission system): A short text summarizing in a single sentence the study (max. 200 characters including spaces). This text is used in conjunction with the titles of papers, hence should be informative and complementary to the title and running title. It should describe the context and significance of the findings for a general readership; it should be written in the present tense and refer to the work in the third person. Author names should not be mentioned.
- By submitting a revision, you attest that you are aware of our payment policies found here: <https://www.life-science-alliance.org/copyright-license-fee>

B. MANUSCRIPT ORGANIZATION AND FORMATTING:

Reviewer #1 (Comments to the Authors (Required)):

The manuscript by Wiese et al. describes a liver epithelial cell-specific knockout of the epigenetic regulator gene KANSL3 in mice. The authors investigate consequences of this knockout, reporting increased MASH scores, collagen deposition and evidence of ductular reaction, as well as transcriptomic changes driven by KANSL3, as determined by RNAseq and CHIPseq. The authors demonstrate that the KANSL3-KO phenotype arises rapidly after birth, as late-stage embryos did not exhibit the same defects as P7 animals. The authors then use scRNAseq of neonates to characterise cellular changes in KANSL3-KO, and again observe increase of cholangiocytes, reduced probability of hepatocyte differentiation, as well as changes in the HSC characteristics of the KO animals. Finally, authors seek to confirm their findings in vitro using organoid models.

While the mouse experiments and sequencing experiments are robust and well controlled, and the work is of potential interest to the field, there are several issues which authors should address before this paper can be published:

Major issues:

- Alb-Cre is not a liver-specific driver, but rather a hepatocyte-specific (or epithelial-specific). This is very visible also in the manuscript's results, as there is still protein present when authors show WB of the KANSL3 protein WT vs KO. Could authors please be more exact in their wording when talking about this specificity?
- What happens at later stages to the KO mice after the initial 3weeks, is there some compensation for the phenotype or do the mice develop problems related to fibrosis and die?
- Authors state: "characterized by the downregulation of key hepatocyte metabolic genes, many of which are directly regulated by KANSL3" but the genes are not shown. Which metabolic genes are meant here by the authors and which metabolic functions? Could the authors show by their CHIP data that KANSL3 binds to the specific genes they mean here?
- DCN is not a specific marker of portal fibroblasts, could authors please check specific genes or show more genes, e.g based on liver cell atlas from Dobie et al. (<https://www.sciencedirect.com/science/article/pii/S2211124719313245>) or Guilliams et al. (<https://www.sciencedirect.com/science/article/pii/S0092867421014811>) ?
- Do the authors see changing proportions of mesenchymal cell types in the KO compared to the WT?
- For the differentiation of WT and KANSL3-KO organoids, the KO seems to just die, rather than fail to differentiate. Could the authors please explore another protocol, e.g. hepatocyte organoids from Hu et al. ([https://www.cell.com/cell/fulltext/S0092-8674\(18\)31505-8](https://www.cell.com/cell/fulltext/S0092-8674(18)31505-8)) or Peng et al. ([https://www.cell.com/cell/fulltext/S0092-8674\(18\)31504-6](https://www.cell.com/cell/fulltext/S0092-8674(18)31504-6)) and show transdifferentiation or more differentiation into cholangiocytes in culture? Another option would be derivation of embryonic cholangiocyte and hepatocyte organoids as in Prior et al. (<https://journals.biologists.com/dev/article/146/12/dev174557/19495/Lgr5-stem-and-progenitor-cells-reside-at-the-apex>) from WT and KO animals to see clear differences in culture of both cell typed.
- Can authors also confirm more/less differentiation using functional assays like cytochrome activity or albumin secretion?

- How can authors really be sure that the imbalance in numbers in scRNAseq is due to the actual imbalance of hepatocytes to BECs? Can authors show FACS plots confirming this? Hepatocytes are at the edge of the 10x chip technology (in size, being very big and often close to the 100um chip), and many could die in the encapsulation.

Minor issues:

- Could authors provide a UMAP representation of WT and KO to compare? It is hard to assess the two conditions as it is.
- "NSL complex" abbreviation should be specified as "non-specific lethal complex" at least once
- Other papers which indicate NLS complex in liver disease should be cited, for e.g. <https://pmc.ncbi.nlm.nih.gov/articles/PMC8934297/>
- In liver sections staining, not only the antibody but also the counter stain for nuclei should be on the figure panel, e.g. as "DAPI"
- Could authors quantify collagen and smooth muscle actin staining to clearly show difference? I agree that the smooth muscle actin differences are there but collagen looks more questionable - the quantification would remove this doubt and improve the manuscript.
- Whole sorting strategy should be shown for FACS in supplemental figures 1i and labels of colours used should also be visible on the axis, together with the protein stained.
- What re the clusters that are not BECs or hepatocytes in fig. 3a-b? This should be described better in text.
- Suppl. Fig. 4d first panel looks different to the other 2 panels, and different to the main UMAP.

Referee Cross-Comments

- While I find the comment of Reviewer 2 about placing KANSL3-KO mice under stress, such as high-fat diet, very valuable and interesting, I am not sure whether this will be within the scope of this manuscript. It would mean a lengthy experimental process and another extensive dataset. Perhaps it is more suitable to a follow up manuscript?
- I find point 1 of Reviewer 3 and the same above point of Reviewer 2 (but in regard to regeneration) very important to investigate in adults upon KO - I would add that perhaps the authors can address it in vitro looking at differentiation and plasticity in hepatocyte and cholangiocyte organoid culture from sorted cells (to remove doubt about cell of origin), but the adult mouse experiment with inducible Cre-KO here and lineage tracing would also be crucial to fully substantiate the claims of the manuscript.
- I find all the other Reviewer 2& 3 comments valid and useful to improve the manuscript.

Reviewer #2 (Comments to the Authors (Required)):

KANSL3 directs transcriptional programs essential for hepatic metabolism and differentiation

In this study, Wiese et al. investigated the role of KANSL3 in the mammalian liver. KANSL3 is a component of the non-specific lethal (NSL) complex, which plays a crucial role in catalyzing histone H4 acetylation at H4K16ac, H4K12ac, H4K8ac, and H4K5ac. To further investigate the role of KANSL3 in the liver, the authors generated liver-specific KANSL3 knockout mice by deleting exons 4 and 5 through breeding floxed KANSL3 mice with Alb-Cre-expressing mice. They reported that KANSL3 liver-KO mice exhibited a normal lifespan and fertility; however, their body weight was reduced at three weeks of age. Macroscopic and microscopic analyses of livers from control and KANSL3 liver-KO mice revealed liver fibrosis and the development of biliary hyperplasia. Bulk RNA-seq of livers from 3-week-old WT and KO animals showed significant gene expression changes, notably the downregulation of key hepatocyte metabolic genes. Additionally, the authors performed scRNA-seq on isolated cell populations enriched for biliary epithelial cells (BECs), hepatic stellate cells (HSCs), and hepatocytes from neonatal (P7) mice. This analysis revealed that the disease phenotype in KANSL3 LKO mice is already established at this early stage. scRNA-seq of neonatal control and KANSL3 LKO livers uncovered a significant shift in liver cellular composition, with a distinct separation between BECs, which were enriched in KANSL3 LKO cells, and hepatocytes, which were predominantly found in control samples. Finally, organoid studies further underscore the crucial role of KANSL3 in the transcriptional activation of genes essential for hepatocyte differentiation. Overall, these findings suggested that KANSL3 LKO hepatic epithelial cells have an impaired ability to differentiate into fully mature hepatocytes, highlighting KANSL3 as a key regulator of liver plasticity.

Major Comments:

- The reviewer is uncertain about the significance of the data due to the lack of long-term effects of KANSL3 loss on liver function and pathology. Have the authors considered placing the knockout mice under stressful conditions? A partial hepatectomy or a high-fat diet (HFD) could be viable options.
- Since KANSL3 catalyzes the acetylation of histone H4 at H4K16ac, H4K12ac, H4K8ac, and H4K5ac, the reviewer questions

why the authors did not examine the acetylation levels of these histone modifications in KANSL3 knockout mice, especially at the genes that exhibited reduced expression.

- In Figure 1m, the authors should clarify how the loss of KANSL3, a known gene activator, resulted in the upregulation of three times as many genes as the number of downregulated genes. Given KANSL3's role in gene activation, one would typically expect the opposite trend.

- It is unclear why differences in body weight are only observed at P21. Can the authors clarify why this time point is unique concerning changes in body weight?

Minor Comments:

Figure 2f,g: Please add a color legend for yellow and grey.

Figure 2h, 2l: The reviewer suggests moving these analyses to the supplementary figures.

Figure 3b: The reviewer requests to make separate plots for each genotype instead of overlapping them.

Figure 3e: The reviewer would like to know what each cluster represents.

Figure 4c: Include a scale bar for images.

Page 9: Correct the reference from Fig. S5E to Fig. S4E.

Reviewer #3 (Comments to the Authors (Required)):

The manuscript by Wiese et al. examines KANSL3, a core component of the NSL complex, in liver development and homeostasis. By deleting floxed *Kansl3* using Albumin-Cre, the authors observe early-onset liver disease, fibrosis, and biliary hyperplasia, suggesting that KANSL3 is essential for hepatocyte differentiation and liver function. They extend these findings to an in vitro organoid system, where KANSL3 deficiency impairs hepatocyte differentiation. Overall, the data convincingly show that KANSL3-mediated regulation is critical for liver development. However, the statements regarding adult liver plasticity and regeneration remain to be substantiated. If confirmed, this study would advance the understanding of how specific epigenetic regulators maintain hepatic function and plasticity.

Major Comments

1. KANSL3's Role in Liver Development vs. Maintenance/Regeneration

The authors provide strong evidence that Albumin-Cre-mediated *Kansl3* deletion leads to severe liver pathology (fibrosis, biliary hyperplasia) by three weeks of age. The results from serum markers and Picrosirius red staining support a pronounced disease phenotype. However, page 12 states that "deleting KANSL3 in 1-week-old mice leads to early-onset liver disease," even though Albumin-Cre is active much earlier, so the data primarily implicate KANSL3 in fetal or neonatal development rather than strictly adult maintenance.

Neonatal murine cells sorted for EPCAM/TSPAN8/CD24 are used in organoid assays to model a "late" *Kansl3* deletion, but these markers also label cells that can give rise to both hepatocytes and cholangiocytes. This assay, therefore, does not exclusively distinguish plasticity from delayed or abnormal differentiation. It remains unclear whether the observed defects in vivo and in vitro reflect a genuine shift in cell fate (plasticity) or simply impaired/delayed maturation.

If the manuscript intends to emphasize a role for KANSL3 in adult hepatocyte maintenance or regeneration, data from inducible deletion in mature hepatocytes (e.g., via tamoxifen-regulated Cre) coupled with adult liver injury or regeneration experiments would clarify whether KANSL3 deficiency disrupts genuine adult plasticity rather than a developmental process. Otherwise, claims regarding adult roles must be adjusted.

2. "Unprecedented Role in Adult Liver Regeneration"

The data convincingly demonstrate KANSL3's importance for liver development and homeostasis, but stating that it provides a "unique mechanism" or "unprecedented role in mediating regeneration of the adult liver" appears overstated. Another NSL subunit, MCRS1, has already been shown to influence adult hepatocyte biology and pathological outcomes (PMID: 34958836). Furthermore, the presented data focus largely on neonates, rather than adult animals.

This point could be addressed by clarifying that KANSL3 contributes to hepatic maintenance and development. If the authors wish to underscore a role in adult regeneration, additional evidence such as adult-onset deletion or injury-based models would be required. Mention of MCRS1 is also relevant because Figure 1B shows that KANSL3 depletion reduces MCRS1, suggesting a broader NSL complex function in liver health.

3. Single-Cell Analysis

The scRNA-seq data convincingly indicate an imbalance between hepatocytes and biliary epithelial cells in neonatal knockout livers, reflecting impaired hepatocyte maturation. However, deducing that KANSL3 is essential for adult plasticity remains challenging without adult deletion or injury-based lineage tracing.

It would be helpful to present KANSL3 expression in different clusters (C1-15) or along the proposed differentiation trajectory (as in Fig. S4E) in order to show how KANSL3 (and possibly other NSL components) is regulated under normal or stressed conditions. Clusters 12, 13, and 15 also exhibit marked differences between genotypes yet receive minimal discussion. More direct comparisons of differentially expressed genes (DEGs) between mutant and wild-type cells in each cluster, rather than just showing cluster-specific marker genes (as in Fig. 2H-I), could further strengthen the single-cell dataset's interpretation.

4. Molecular Mechanism: NSL Complex and Histone Acetylation

The ChIP-seq analysis for KANSL3 in wild-type livers, overlaid with DEG promoters, suggests direct regulation of downregulated genes. However, selection criteria (e.g., 490 genes vs. all 3148 downregulated genes) require clarification. The NSL complex is

known to mediate histone H4 acetylation, so measuring relevant histone marks (H4K16ac, H4K12ac, etc.) after Katsl3 deletion would offer more mechanistic insight and confirm that NSL-mediated acetylation is altered in specific gene targets in the liver.

5. Additional Functional Readouts

The transcriptomic data highlight disruptions in lipid and steroid metabolism. Oil Red O staining at E17.5, P7, and P21 might reveal the timing of any lipid accumulation or liver metabolic dysfunction. Staining for and quantifying canonical hepatocyte markers, such as HNF4A, at multiple stages would further support the conclusions about hepatocyte-to-BEC identity shifts and their timing in development.

6. Early or Transient Cell Death?

The text indicates that apoptosis is not elevated at P7, suggesting that cell death is unlikely to fully explain the altered cell compositions. However, cell death could have occurred at an earlier stage. More comprehensive time-course analyses or additional histological measures of apoptosis might help corroborate whether the effect on hepatocyte numbers is indeed due to differentiation defects rather than early or transient cell death.

Minor Comments

- Figure 2F/G: A clearer labeling of yellow vs grey clusters (e.g., highlighting which cells are knockout vs. control) would improve readability (and is nicely done in the other figures).
- Abbreviations: Spell out "NSL complex" and similar terms upon first mention.
- Figure 1B: Include quantitative Western blot data rather than relying solely on representative images.
- Collagen/ α -SMA: Include quantifications for fibrotic markers (e.g., Figure S1G,H).
- References to Cholangiocyte-Derived Hepatocyte Regeneration: The Forbes lab (PMID: 28700576) showed that cholangiocyte contribution to new hepatocytes requires significant hepatocyte impairment, implying that such BEC-to-hepatocyte transitions do not occur under normal conditions. The discussion might acknowledge that plasticity typically requires acute or chronic injury with a block in hepatocyte-mediated regeneration.

We thank all reviewers for their valuable feedback. Please find our point-by-point reply below. Our reply is marked in blue.

Summary of key data added upon revision:

1. Fetal liver organoids:

Kansl3 iKO Hep-like fetal organoids show growth defects, reduced Cyp3a4 activity, and impaired Albumin secretion, indicating KANSL3's essential role in hepatocyte differentiation during fetal development. (Fig. 6.)

2. Validation of hepatocyte and BEC imbalance in vivo:

New mouse model crossing mTmG reporter with Kansl3 LKO mice confirms an imbalance between hepatocytes and BECs in neonates (Supplemental Fig. 5N-P) and identifies increased apoptosis in Kansl3 LKO hepatocytes (Fig. 3I).

3. KANSL3 regulates hepatic metabolic genes via histone acetylation:

- Rapid degradation of KANSL3 in hepatocyte cell line Hepa1-6 cells reveals overlaps in gene expression changes across bulk liver, neonatal hepatocytes, and BEC-to-Hep organoids (Supplemental Fig. 3A-D).
- Ectopic expression of full length KANSL3 rescues down-regulation of gene expression in dTAG13 treated dTAG-Kansl3 Hepa-16 cells (Supplemental Fig. 3E-G).
- H4K16ac ChIP-seq in dTAG-Kansl3 Hepa-16 cells shows decreased H4K16ac at down-regulated genes, and TSA treatment rescues these gene expression changes, confirming acetylation-dependent regulation by the NSL complex (Supplemental Fig. 3H-J).

4. KANSL3-mediated gene down-regulation:

- scRNA-seq differential expression analysis shows down-regulation of genes related to hepatocyte metabolic functions (fatty acid metabolism, response to xenobiotic stimulus, mitochondrial function) and cilia function in hepatocyte clusters (Fig. 3H).
- KANSL3 is enriched at TSS of genes down-regulated in *Kansl3* LKO P21 livers, and enriched for metabolic functions 'fatty acid metabolism', 'steroid metabolism' and 'response to xenobiotic stimulus' over a random set of genes (Fig. 2G, Supplemental Fig. 2F).

5. Metabolic Defects in vivo:

RNA-seq analysis detects lipid and steroid metabolic defects in Kansl3 LKO liver, validated by Oil Red O and total cholesterol assays (Fig. 2D-E).

6. Long-term Effects of Kansl3 LKO:

Old Kansl3 LKO mice exhibit signs of premalignant growth, underscoring long-term effects of KANSL3 depletion during development (Supplemental Fig. 4).

Reviewer #1 (Comments to the Authors (Required)):

The manuscript by Wiese et al. describes a liver epithelial cell-specific knockout of the epigenetic regulator gene KANSL3 in mice. The authors investigate consequences of this knockout, reporting increased MASH scores, collagen deposition and evidence of ductular reaction, as well as transcriptomic changes driven by KANSL3, as determined by RNAseq and CHIPseq. The authors demonstrate that the KANSL3-KO phenotype arises rapidly after birth, as late-stage embryos did not exhibit the same defects as P7 animals. The authors then use scRNAseq of neonates to characterise cellular changes in KANSL3-KO, and again observe increase of cholangiocytes, reduced probability of hepatocyte differentiation, as well as changes in the HSC characteristics of the KO animals. Finally, authors seek to confirm their findings in vitro using organoid models.

While the mouse experiments and sequencing experiments are robust and well controlled, and the work is of potential interest to the field, there are several issues which authors should address before this paper can be published:

We are happy to read this reviewer for considering our work as “robust and well controlled” and recognizing it as “of potential interest to the field”.

Major issues:

- Alb-Cre is not a liver-specific driver, but rather a hepatocyte-specific (or epithelial-specific). This is very visible also in the manuscript's results, as there is still protein present when authors show WB of the KANSL3 protein WT vs KO. Could authors please be more exact in their wording when talking about this specificity?

We thank the reviewer for this comment. We clarified the specificity of the Alb-Cre in the hepatocyte as well as the BEC compartment in the text. We also added a comment clarifying the residual expression of KANSL3 protein and RNA levels in bulk tissues due to non-epithelial cell types.

- What happens at later stages to the KO mice after the initial 3 weeks, is there some compensation for the phenotype or do the mice develop problems related to fibrosis and die?

This is indeed an important question. KANSL3 LKO mice survive, are fertile, and show normal lifespan, indicating a compensation mechanism that allows the liver to overcome the initial damage presented at a young age. Notably, KANSL3 protein levels partially recover over time, suggesting counterselection against KANSL3-deficient cells during liver regeneration. However, at old age (90 weeks), while fibrosis and liver function defects ameliorate, the mice develop premalignant changes in the tissue, indicating that the KANSL3 deletion early in development

has long-term consequences. Exact mechanisms of how this premalignant state arises require a tighter time course analysis and remain to be elucidated in the future. We have added this data in Supplemental Fig. S4.

- Authors state: "characterized by the down-regulation of key hepatocyte metabolic genes, many of which are directly regulated by KANSL3" but the genes are not shown. Which metabolic genes are meant here by the authors and which metabolic functions? Could the authors show by their CHIP data that KANSL3 binds to the specific genes they mean here?

The deletion of KANSL3 in hepatocytes leads to the down-regulation of 3,148 genes, with 490 genes changing expression with a log2 fold change of <-1. Notably, down-regulated genes are enriched for fatty acid and steroid metabolism pathways, as well as xenobiotic response processes (Fig. 2B,C, Supplemental Fig. S2C, heatmaps list gene names), indicating that transcriptional programs underpinning core hepatic functions are disrupted in KANSL3 LKO livers. Consistent with these gene expression changes, we observed increased lipid accumulation (Fig. 2D) and total cholesterol levels (Fig. 2E) in the liver.

Our analysis revealed that KANSL3 is specifically enriched at the transcription start sites (TSS) of genes related to fatty acid metabolism (e.g., *Mecr* and *Acms2*), steroid metabolism (e.g., *Sreb1* and *Hsd11b1*), and xenobiotic response (e.g., *Xbp1* and *Blmh*) compared to a random set of genes (Fig. 2G, Supplemental Fig. S2F,G). In contrast, up-regulated genes related to extracellular matrix (ECM) organization or response to wound healing show no enrichment of KANSL3 over random genes (Supplemental Fig. S2F), suggesting a secondary response to the loss of KANSL3 in hepatocytes in the tissue microenvironment. Overall, our results demonstrate that KANSL3 plays a crucial role in regulating key pathways for hepatocyte function.

- DCN is not a specific marker of portal fibroblasts, could authors please check specific genes or show more genes, e.g based on liver cell atlas from Dobie et al. (<https://www.sciencedirect.com/science/article/pii/S2211124719313245>) or Guilliams et al. (<https://www.sciencedirect.com/science/article/pii/S0092867421014811>)?

We thank the reviewer for their input. Based on the suggested single cell atlas publications, we have re-annotated the stromal cell populations in clusters C3, 8,10 and 12 (Fig. 3D, Supplemental Fig. 6A). All populations express to some degree of the hepatic stellate cell marker *Acta2*, however, we updated the annotation as follows:

Cluster 3: We agree that the annotation of "portal fibroblasts" for Cluster 3 was incorrect. We have now re-annotated cluster 3 into "fibroblasts" as Cluster 3 expresses classical markers of fibroblasts (for example: *Dcn*, *Dpt*, *Col1a1* *Col1a2*, *Col3a1*). Further subclassification of cluster 3 remains challenging as it is a heterogeneous population expressing both markers of central vein fibroblasts (*Dpt*, *Gsn*, *Atp1a2*) and bile duct fibroblasts (*Mfab4*, *Svep1*).

Cluster 8: Hepatic stellate cells (HSCs)/Vascular smooth muscle cells (VSMCs) as cells express both the marker genes for HSCs (HSCs: *Acta2*, *Lox*, *Reln*, *Sox4*, *Ecm1*, *Loxl1*) as well as for VSMCs (*Tagln*, *Cnn1* or *Myh11*).

Cluster 10: HSC/Myofibroblasts as they as they express both the marker genes for HSCs (*Acta2*, *Lox*, *Reln*, *Sox4*, *Ecm1*, *Loxl1*) with high levels of *Col1a1*, *Col1a2*, *Col3a1*, *Lox* and *Acta2*.

Cluster 12: HSCs with a preference for portal vein HSCs, less active as they express lower levels of *Acta2/Lox*.

The resolution of clustering employed here does not allow further subclassification of the stromal cell populations (C3,8,10,11,12).

- Do the authors see changing proportions of mesenchymal cell types in the KO compared to the WT?

We have added a description of the changes in the stromal compartment in the text. The data concerning these cell types are now contained in Supplemental Fig. S6A-E. Stromal cells are found in the clusters C3, C8, C10, C11 and C12. For clusters 3 and 10 we observed a reduction of cells in the Kansl3 LKO mice (Supplemental Fig. S6D). Cluster 8 showed a mild increase in Kansl3 LKO mice, while a more pronounced increase in cell number was observed for cluster 12. While the numbers of cells within the different stromal cell populations overall do not indicate a significant shift as observed in the epithelial cell populations, differential expression analysis between controls and Kansl3 LKO cells combining all stromal cells showed an upregulation of pathways related to fibrosis and tissue reorganization such as collagen catabolism, keratan sulfate degradation or disease of glycosylation (Supplemental Fig. S6E). These results are in line with the increased fibrosis observed in vivo.

- For the differentiation of WT and KANSL3-KO organoids, the KO seems to just die, rather than fail to differentiate.

We conducted Casp7/3 apoptosis assays on day 15 of differentiation and found only a mild, non-significant increase in apoptosis (Fig. 5D). Correspondingly, apoptotic genes were not prominently upregulated during differentiation (Fig. 5H). As noted by the reviewer, Kansl3 iKO organoids were smaller and morphologically altered compared to controls on day 15 (Fig. 5B) and indeed we observed a significant proliferation defect upon addition of 4OHT (Fig. 5C).

In contrast to the organoid model, in vivo results added upon revision revealed an upregulation of apoptotic genes (Fig. 3H) and increased Annexin V positive hepatocytes in P7 mTmG Kansl3 LKO when compared to mTmG control mice (Fig. 3G,I) with no predominant proliferation defects at the gene expression level (Fig. 3H).

Our findings reveal a discrepancy between in vivo and in vitro models: While in isolation, loss of KANSL3 may result in reduced proliferation, in a tissue context KANSL3 depleted cells undergo

increased apoptosis potentially due to crosstalk with other tissue resident cells such as immune cells. Despite this, we showed that KANSL3 regulates hepatic metabolic genes across different models (in P21 bulk liver: Fig. 2, Supplemental Fig. S2C,F,G; in P7 hepatocytes: Fig. 3H; in organoids Fig. 5F-I, Supplemental Fig. S8&9; in Hepa1-6 cells: Supplemental Fig. S3B-D).

In addition, genes related to mitochondrial function and cilia biogenesis/cell motility were down-regulated both in vivo and in vitro, suggesting that KANSL3 KO impacts multiple cellular functions, which may contribute to the observed phenotypes in vivo and in vitro.

Could the authors please explore another protocol, e.g. hepatocyte organoids from Hu et al. ([https://www.cell.com/cell/fulltext/S0092-8674\(18\)31505-8](https://www.cell.com/cell/fulltext/S0092-8674(18)31505-8)) or Peng et al. ([https://www.cell.com/cell/fulltext/S0092-8674\(18\)31504-6](https://www.cell.com/cell/fulltext/S0092-8674(18)31504-6)) and show transdifferentiation or more differentiation into cholangiocytes in culture? Another option would be derivation of embryonic cholangiocyte and hepatocyte organoids as in Prior et al. (<https://journals.biologists.com/dev/article/146/12/dev174557/19495/Lgr5-stem-and-progenitor-cells-reside-at-the-apex>) from WT and KO animals to see clear differences in culture of both cell typed.

As requested by the reviewer, we employed an additional organoid protocol. The early manifestation of the phenotype in the mouse suggested defects during fetal liver development. Hence, we generated fetal liver (FL) organoids from *Kansl3^{fl/fl}* and *Kansl3^{fl/fl}* and *Cag-CreERT2^{T/+};Kansl3^{fl/fl}* E12.5 embryos following the method outlined by Prior et al. (PMID: 31142540). These unsorted fetal liver cells were cultured for 14 days to differentiate into hepatocyte-like FL organoids (Fig. 6A).

Kansl3 knockout was induced on day 3 of differentiation, leading to observable growth defects in organoids (Fig. 6B), similar to BEC-to-Hep organoids (Fig. 5B,C). While hepatocyte markers *Alb*, *Hnf4a*, and *Cyp3a11* remained unchanged, mitochondrial genes such as *Mp11* and *Ndufb3* were significantly down-regulated, indicating consistency across models (Fig. 6C, Supplemental Fig. S3D). Notably, *Kansl3* iKO organoids exhibited reduced *Cyp3a4* activity (Fig. 6D) and Albumin secretion (Fig. 6E), revealing partially impaired hepatocyte function.

Despite initial intentions to differentiate the fetal liver cells into Hep-like and BEC-like organoids, technical challenges prevented BEC-like organoid growth, even in the absence of 4OHT. Therefore, we could not assess the effect of KANSL3 loss on BEC differentiation.

However, expression of BEC markers *Krt19*, *Epcam* and *Sox9* was unchanged in *Kansl3* iKO Hep-like organoids (see below, significance was scored by Mann-Whitney test) suggesting that BEC specific gene expression programs were not activated in these organoids. In the case of BEC-to-Hep organoids we hypothesize that the absence of KANSL3 hinders the proper activation of essential gene expression programs for hepatocyte differentiation, causing BEC progenitor organoids to maintain their BEC-like state, as evidenced by the upregulation of BEC-specific genes in *Kansl3* iKO BEC-to-Hep organoids (Fig. 5G).

- Can authors also confirm more/less differentiation using functional assays like cytochrome activity or albumin secretion?

We demonstrate that *Kansl3* iKO Hep-like FL organoids showed decreased Cyp3a4 activity (Fig. 6D) and Albumin secretion (Fig. 6E) on day 14 of differentiation suggesting partially impaired hepatocyte function upon KANSL3 depletion. See above for a discussion of these results.

- How can authors really be sure that the imbalance in numbers in scRNAseq is due to the actual imbalance of hepatocytes to BECs? Can authors show FACS plots confirming this? Hepatocytes are at the edge of the 10x chip technology (in size, being very big and often close to the 100um chip), and many could die in the encapsulation.

We appreciate the reviewer's insightful comment regarding the 10x chip technology's limitations with large cells like hepatocytes, which indeed may have led to a bias towards smaller cells in our scRNA-seq data. While polynucleated hepatocytes may be underrepresented due to encapsulation challenges, we ensured that the technique was consistently applied to both control and *Kansl3* LKO liver cells, allowing internally controlled comparisons. This comparison revealed a reduction in cells with hepatocyte transcriptional profiles (clusters 0 and 6) in *Kansl3* LKO mice versus control mice.

To further validate the imbalance between hepatocytes and BECs following *Kansl3* KO, we crossed KANSL3 LKO mice with mTmG reporter mice to GFP-label AlbCre active cells. Using FACS analysis on P7 liver cells, we confirmed a decrease in GFP+ EpCAM- hepatocytes and an increase in total EpCAM+ BECs in KANSL3 LKO mice, consistent with scRNA-seq findings

(Supplemental Fig. S5N-P). The total BEC population was made up of GFP+ (~65%) and GFP-BECs (~35%), reflecting the heterogeneity within the BEC compartment.

It is important to point out here that, while the overall viability of cells analyzed was high (>90%), the digestion protocol used for neonatal livers is relatively harsh for hepatocytes, resulting in a lower-than-expected yield of GFP+ EpCAM- hepatocytes (~ 10-15% of viable cells) in controls. This is due to the challenges associated with perfusing neonatal animals. However, our results show that *Kansl3* LKO hepatocytes are even more fragile than controls, with a significantly reduced yield of viable cells, indicating that the loss of KANSL3 reduces the fitness of hepatocytes.

Minor issues:

- Could authors provide a UMAP representation of WT and KO to compare? It is hard to assess the two conditions as it is.

As requested, a Control and *Kansl3* LKO cell identity UMAP of the epithelial zoom is shown in Fig. 4D.

- "NSL complex" abbreviation should be specified as "non-specific lethal complex" at least once
Updated.

- Other papers which indicate NLS complex in liver disease should be cited, for e.g.
<https://pmc.ncbi.nlm.nih.gov/articles/PMC8934297/>

We have updated the discussion accordingly.

- In liver sections staining, not only the antibody but also the counter stain for nuclei should be on the figure panel, e.g. as "DAPI"

The DAPI channel was added as requested (Fig. 1I-K).

- Could authors quantify collagen and smooth muscle actin staining to clearly show difference? I agree that the smooth muscle actin differences are there but collagen looks more questionable - the quantification would remove this doubt and improve the manuscript.

We thank the reviewer for this comment. We have repeated the experiment and quantified both the area of α -SMA and COLLAGEN staining, both showing increased staining in *Kansl3* LKO samples (Fig. 1I/J). In addition, we performed co-staining of both α -SMA and COLLAGEN to qualitatively show the expected overlap of both proteins (Supplemental Fig. S1G).

- Whole sorting strategy should be shown for FACS in supplemental figures 1i and labels of colours used should also be visible on the axis, together with the protein stained.

The FACS plots and labels have been updated, now located in Supplemental Fig. S1H.

- What are the clusters that are not BECs or hepatocytes in fig. 3a-b? This should be described better in text.

Due to the similarity between BECs (C0, C4, C5, C10, C12), hepatocytes (C1, C2, C3, C6) and intermediate clusters expressing markers for both cell types (C7, C3) (Fig. 4A-C, Supplemental Fig. 6F, Supplemental Table S3), and the resolution of our scRNA-seq data, a complete annotation of all clusters remains challenging. However, we have added more details to the clusters where applicable.

BEC cluster C5 showed an inflammatory signature (*Ccl2*, *Cxcl1*, *Cxcl2* and *Tnf* expression), while cluster C10 exhibited a more proliferative signature (*Pcna*, *Mki67*, *Cks1b* and *Cdc20* expression). Clusters C0, C4, and C12 expressed markers of BEC progenitors (*Cd24*, *Prom1*, *St14*) (Supplemental Table S3). Interestingly, an expression gradient of the BEC marker *Tspan8* was detected across all BECs, with highest expression in C12 and C4, medium expression in C0 and C10, and low expression in cluster C5 (Supplemental Fig. S6F). Hepatocyte clusters C1, C2 and C6 all express high levels of core metabolic genes, while C6 also showed a proliferative signature (*Cdc20*, *Cdk1*, *Cdkn3*) (Fig. 4B, Supplemental Table S3). Remarkably, *Kansl3* LKO mice showed a significant reduction in hepatocyte clusters C1 and C6, and an almost complete absence of the intermediate clusters C7 and C3 (Fig. 4D, Supplemental Table S3), while cluster C2 showed equal numbers (Fig. 4D, Supplemental Table S3). Within the BECs compartment, an increase in clusters C0, C4, and C10 was observed, accompanied by a decrease in cluster C5 (Fig. 4D, Supplemental Table S3).

In addition, we have annotated non-hepatic epithelial cell clusters C8, C9, C11, C13, and C14, which predominantly represent background (non-hepatic epithelial cell) populations. We deliberately included these cells to maintain the integrity of our zoomed-in dataset and avoid overly strict boundaries.

All marker genes according to Seurat as well as the composition of each cluster are now available in Supplemental Table S3.

- Suppl. Fig. 4d first panel looks different to the other 2 panels, and different to the main UMAP. We thank this reviewer for spotting this inconsistency. We have now removed the UMAP for *Col1a1* because we believe that *Des* and *Acta2* are sufficient to annotate these cells as HSCs. (now Supplemental Fig. S6H)

Referee Cross-Comments

- While I find the comment of Reviewer 2 about placing KANSL3-KO mice under stress, such as high-fat diet, very valuable and interesting, I am not sure whether this will be within the scope of this manuscript. It would mean a lengthy experimental process and another extensive dataset. Perhaps it is more suitable to a follow up manuscript?

Exploring a dietary intervention in *Kansl3* LKO mice would indeed be intriguing for understanding KANSL3's role in metabolic regulation and liver regeneration. However, we concur with the reviewer that these experiments are beyond the scope of the current manuscript and would be more appropriate for a follow-up study.

- I find point 1 of Reviewer 3 and the same above point of Reviewer 2 (but in regard to regeneration) very important to investigate in adults upon KO - I would add that perhaps the authors can address it in vitro looking at differentiation and plasticity in hepatocyte and cholangiocyte organoid culture from sorted cells (to remove doubt about cell of origin), but the adult mouse experiment with inducible Cre-KO here and lineage tracing would also be crucial to fully substantiate the claims of the manuscript.

We agree with the reviewers on the importance of employing an inducible Alb-Cre driven KO model coupled with lineage tracing to study KANSL3's role in liver regeneration. Unfortunately, developing new mouse models and obtaining the necessary animal ethics license in Germany is a lengthy process, exceeding the current revision timeline. Instead, we have moderated our claims regarding KANSL3's role in liver regeneration.

Acknowledging our current mouse model's limitations, particularly regarding the gradual Alb expression and KANSL3 depletion along the BEC-to-Hep trajectory, we intentionally sorted BEC progenitor cells using scRNA-seq predictions to validate our in vivo results. Organoid formation assays using progenitor cells (EpCAM high, Tspan8 high and CD24 positive), Tspan8 medium (EpCAM high, Tspan8 medium, CD24 negative) and Tspan8 low (EpCAM high, Tspan8 low, CD24 negative) from P7 control mice showed a decreased organoid formation potential with reducing Tspan8 levels (Supplemental Fig. S7B/C), affirming the predicted progenitor cell population. Cells isolated from KANSL3 LKO mice exhibited significantly lower organoid-forming potential compared to controls, suggesting defects in these cells (Supplemental Fig. S7B/C).

Using organoids from *KANSL3^{fl/fl}* and *KANSL3^{fl/fl}* and *Cag-CreERT2^{T/+};KANSL3^{fl/fl}* mice we could deplete KANSL3 in a relatively homogenous cell population (in contrast to the relatively undefined onset of Alb-Cre activity) to test the effect on BEC-to-Hep differentiation. Our results showed that essential gene expression programs for hepatocyte differentiation were not induced without KANSL3, potentially causing BEC progenitor organoids to persist in a more BEC-like state. This was evidenced by the upregulation of BEC-specific genes in KANSL3 iKO BEC-to-Hep organoids (Fig. 5G). However, further investigation is needed to understand whether KANSL3 is also involved in the exit of BEC cellular identity during BEC-to-Hep transdifferentiation.

To support our findings, we utilized an additional organoid model. KANSL3 iKO fetal liver Hep-like organoids exhibited compromised differentiation, akin to BEC-to-Hep organoids. Our results emphasize KANSL3's crucial role in hepatocyte differentiation and function, both during transdifferentiation and during hepatoblast differentiation into hepatocytes. We observed that KANSL3 loss affects differentiation pathways in both Hep-like FL and BEC-to-Hep organoids, however, with different gene sets may be involved. We acknowledge that more work is needed to understand the exact mechanisms.

- I find all the other Reviewer 2& 3 comments valid and useful to improve the manuscript.

Reviewer #2 (Comments to the Authors (Required)):

KANSL3 directs transcriptional programs essential for hepatic metabolism and differentiation

In this study, Wiese et al. investigated the role of KANSL3 in the mammalian liver. KANSL3 is a component of the non-specific lethal (NSL) complex, which plays a crucial role in catalyzing histone H4 acetylation at H4K16ac, H4K12ac, H4K8ac, and H4K5ac. To further investigate the role of KANSL3 in the liver, the authors generated liver-specific *Kansl3* knockout mice by deleting exons 4 and 5 through breeding floxed *Kansl3* mice with Alb-Cre-expressing mice. They reported that *Kansl3* liver-KO mice exhibited a normal lifespan and fertility; however, their body weight was reduced at three weeks of age. Macroscopic and microscopic analyses of livers from control and KANSL3 liver-KO mice revealed liver fibrosis and the development of biliary hyperplasia. Bulk RNA-seq of livers from 3-week-old WT and KO animals showed significant gene expression changes, notably the down-regulation of key hepatocyte metabolic genes. Additionally, the authors performed scRNA-seq on isolated cell populations enriched for biliary epithelial cells (BECs), hepatic stellate cells (HSCs), and hepatocytes from neonatal (P7) mice. This analysis revealed that the disease phenotype in *Kansl3* LKO mice is already established at this early stage. scRNA-seq of neonatal control and *Kansl3* LKO livers uncovered a significant shift in liver cellular composition, with a distinct separation between BECs, which were enriched in *Kansl3* LKO cells, and hepatocytes, which were predominantly found in control samples. Finally, organoid studies further underscore the crucial role of KANSL3 in the transcriptional activation of genes essential for hepatocyte differentiation. Overall, these findings suggested that *Kansl3* LKO hepatic epithelial cells have an impaired ability to differentiate into fully mature hepatocytes, highlighting KANSL3 as a key regulator of liver plasticity.

Major Comments:

- The reviewer is uncertain about the significance of the data due to the lack of long-term effects of *Kansl3* loss on liver function and pathology. Have the authors considered placing the knockout mice under stressful conditions? A partial hepatectomy or a high-fat diet (HFD) could be viable options.

We agree with the reviewer's suggestion that applying stress to the liver in adult *Kansl3* LKO mice through a dietary challenge or acute/chronic hepatocyte damage would offer valuable insights into KANSL3's role in metabolic regulation and liver regeneration. However, as already noted by Reviewer 1, we believe that such experiments are beyond the scope of this manuscript.

Nonetheless, we have explored the effects of KANSL3 loss in hepatocytes in adult animals. KANSL3 LKO mice survive, are fertile and show normal lifespan indicating a compensation mechanism. This suggests that the liver does overcome the initial damage presented at a young age. Interestingly, at old age (90 weeks), while fibrosis and liver function defects ameliorate, the mice develop premalignant changes in the tissue indicating that the KANSL3 deletion early in development of the tissue has consequences later in life. Exact mechanisms of how this premalignant state arises require a tighter time course analysis and remain to be elucidated in the future. We have added this data in Supplemental Fig. S4.

- Since KANSL3 catalyzes the acetylation of histone H4 at H4K16ac, H4K12ac, H4K8ac, and H4K5ac, the reviewer questions why the authors did not examine the acetylation levels of these histone modifications in KANSL3 knockout mice, especially at the genes that exhibited reduced expression.

We appreciate the reviewer's questions regarding histone acetylation and its regulatory role. To study KANSL3's transcriptional regulation mechanisms and avoid limitations of bulk tissue analyses, we developed an in vitro hepatocyte model using the dTAG system in the murine Hepa1-6 cell line. This allowed us to examine early transcriptomic changes following acute KANSL3 depletion. RNA-seq analysis in Hepa 1-6 HA-dTAG-KANSL3 cells, conducted 24 hours after KANSL3 degradation, revealed significant transcriptomic changes, with 2,202 genes down-regulated and 1,762 upregulated compared to control treatment (Supplemental Fig. 3B, Table S2). down-regulated genes were enriched in mitochondrial functions, lipid metabolism, and cilia-related terms (Supplemental Fig. 3C,D), indicating parallels between acute KANSL3 depletion in cells and long-term loss in liver tissue (Fig. 2B). Notably, these expression changes were rescued by ectopic expression of a FLAG-tagged human full-length KANSL3 protein (Supplemental Fig. 3E-G), indicating direct regulation by KANSL3.

KANSL3 ChIP-qPCR confirmed its binding to the promoters, rather than the 3'-ends, of target genes like *Mrp11* and *Fah*, which was abolished post-dTAG-13 treatment (Supplemental Fig. 3H). ChIP-seq analysis showed reduced H4K16ac levels at KANSL3 target genes following degradation, highlighting histone acetylation-dependent regulation by the NSL complex (Supplemental Fig. 3I). Furthermore, treatment of dTAG-13 treated cells with the HDAC inhibitor Tricostatin A (TSA) rescued gene expression of KANSL3 target genes *Mrp11*, *Fah*, *Mrp55*, and *Nduf3b* (Supplemental Fig. S3J). While expression returned to control levels in the case of *Mrp11*, expression of *Fah*, *Mrp55*, and *Nduf3b* exceeded control levels. Taken together this data supports acetylation-dependent gene regulation by KANSL3.

While we aimed to investigate the effect of KANSL3 degradation on other NSL-complex associated histone H4 acetylation marks (H4K5ac and H4K8ac), technical challenges prevented us from doing so within the revision timeframe. However, given the known preference of NSL depletion for K5 and K8 residues in other cellular systems (PMID: 33657400), we anticipate that the observed changes would be even more pronounced.

- In Figure 1m, the authors should clarify how the loss of KANSL3, a known gene activator, resulted in the upregulation of three times as many genes as the number of down-regulated genes. Given KANSL3's role in gene activation, one would typically expect the opposite trend.

In our analysis of bulk liver tissue with criteria of $FDR < 0.05$ and using a \log_2FC cutoff of < -1 for down-regulation and > 1 for upregulation, we identified 490 down-regulated and 1560 upregulated genes. As the reviewer noted, KANSL3 is a known transcriptional activator as part of the NSL complex, suggesting initial depletion would typically lead to transcriptional down-regulation. However, we strongly believe that the observed upregulation of genes represents a secondary response to KANSL3 loss, driven by the pronounced fibrosis, ductular reaction, and inflammation phenotypes in P21 animals. Many of the upregulated genes are not even

expressed in hepatocytes/hepatic epithelial cells, suggesting that they are not directly regulated by KANSL3.

KANSL3 ChIP-seq analysis in control liver tissue showed enrichment at the transcription start sites (TSS) of down-regulated genes, more than at up-regulated ones (Fig. 2F). Enhanced binding of KANSL3 was seen at TSS of genes involved in fatty acid metabolism, such as *Mecr* and *Acms2*, steroid metabolism like *Srebf1* and *Hsd11b1* (Fig. 2G), and response to xenobiotic stimuli such as *Xbp1* and *Blmh* (Supplemental Fig. S2F,G), further underlining KANSL3's role in controlling pathways of hepatocyte function.

Conversely, upregulated genes for example related to ECM organization or response to wound healing showed no KANSL3 enrichment at their TSS compared to a random gene set (Supplemental Fig. S2F), suggesting that their regulation is independent of KANSL3 and likely a secondary response to KANSL3 depletion within the hepatocyte tissue microenvironment.

- It is unclear why differences in body weight are only observed at P21. Can the authors clarify why this time point is unique concerning changes in body weight?

We appreciate the reviewer's question. The observed differences in body weight at P21 are intriguing, and we believe that the liver undergoes a type of crisis at this point. While no marked changes were observed at E17.5 at the tissue level, fibrosis and biliary hyperplasia develop in neonates (or within the first week of life) and worsen over time. The transition to solid food around weaning also falls into this time window and may contribute to this phenotype. Animals with hepatocyte specific KO of *Ogt*, another NSL complex member, develop moderate liver disease signs at 4 weeks of age progressing to severe liver disease at 8 weeks. Interestingly, a ketogenic (low carb/fat) diet at weaning can prevent the disease onset (PMID: **38298740**). While the progression of the disease is slower in the case of *Ogt* LKO compared to *Kansl3* LKO, it would be interesting to investigate whether a ketogenic diet switch could also ameliorate the *Kansl3* LKO phenotype.

At P21, the liver does not regenerate through hepatocyte proliferation, as evidenced by a lack of increased liver/body weight ratio (Supplemental Fig. S1E). However, by 6-12 weeks, liver/body weight ratios increase, suggesting that the liver is able to overcome the initial damage presented at a young age. This is why we chose to investigate the molecular changes at early timepoints. As noted by this reviewer, the Alb-Cre mouse model has its limitations, and an inducible system would be beneficial for studying KANSL3 function at different stages of liver development and in adult liver homeostasis. However, this is beyond the scope of our current manuscript.

To elucidate the long-term consequences of KANSL3 deletion, we have added data from old (90-week-old) *Kansl3* LKO animals. Old *Kansl3* LKO mice survive, are fertile, and show normal lifespan, indicating a compensation mechanism. Notably, while fibrosis and liver function defects improve, premalignant changes develop in the tissue, suggesting long-term consequences of early KANSL3 deletion. The exact mechanisms of this premalignant state require further investigation. We have added this new data in Supplemental Fig. S4.

Minor Comments:

Figure 2f,g: Please add a color legend for yellow and grey.

We apologize for this unclarity. The labels (grey: control, yellow: KO) have now been added to the figure. The figures are now located in Fig. 3E/F.

Figure 2h, 2l: The reviewer suggests moving these analyses to the supplementary figures.

The analyses were moved to Supplemental Fig. S5K/L.

Figure 3b: The reviewer requests to make separate plots for each genotype instead of overlapping them.

The updated figure is shown in Fig. 4D.

Figure 3e: The reviewer would like to know what each cluster represents.

We appreciate the reviewer's request for clarification on the clusters now in Fig. 4E. We note that the Seurat clustering of the epithelial zoom (Fig. 4A) is similar to the RaceID clustering necessary for the pseudotime analysis (Fig. 4E). As a result, we have focused on providing additional information on the clusters depicted in Fig. 4A.

Due to the similarity between BECs (C0, C4, C5, C10, C12), hepatocytes (C1, C2, C3, C6) and intermediate clusters expressing markers for both cell types (C7, C3) (Fig. 4A-C, Supplemental Fig. 6F, Supplemental Table S3), and the resolution of our scRNA-seq data, a complete annotation of all clusters remains challenging. However, we have added more details to the clusters where applicable.

BEC cluster C5 showed an inflammatory signature (*Ccl2*, *Cxcl1*, *Cxcl2* and *Tnf* expression), while cluster C10 exhibited a more proliferative signature (*Pcna*, *Mki67*, *Cks1b* and *Cdc20* expression). Clusters C0, C4, and C12 expressed markers of BEC progenitors (*Cd24*, *Prom1*, *St14*) (Supplemental Table S3). Interestingly, an expression gradient of the BEC marker *Tspan8* was detected across all BECs, with highest expression in C12 and C4, medium expression in C0 and C10, and low expression in cluster C5 (Supplemental Fig. S6F). Hepatocyte clusters C1, C2 and C6 all express high levels of core metabolic genes, while C6 also showed a proliferative signature (*Cdc20*, *Cdk1*, *Cdkn3*) (Fig. 4B, Supplemental Table S3). Remarkably, *Kansl3* LKO mice showed a significant reduction in hepatocyte clusters C1 and C6, and an almost complete absence of the intermediate clusters C7 and C3 (Fig. 4D, Supplemental Table S3), while cluster C2 showed equal numbers (Fig. 4D, Supplemental Table S3). Within the BECs compartment, an increase in clusters C0, C4, and C10 was observed, accompanied by a decrease in cluster C5 (Fig. 4D, Supplemental Table S3).

In addition, we have annotated non-hepatic epithelial cell clusters C8, C9, C11, C13, and C14, which predominantly represent background (non-hepatic epithelial cell) populations. We deliberately included these cells to maintain the integrity of our zoomed-in dataset and avoid overly strict boundaries.

All marker genes according to Seurat as well as the composition of each cluster are now available in Supplemental Table S3.

Figure 4c: Include a scale bar for images.

We thank the reviewer for spotting this inconsistency. Scale bars have now been added to the panels (Fig. 5B).

Page 9: Correct the reference from Fig. S5E to Fig. S4E.

We thank the reviewer for spotting this inconsistency. The reference has been updated.

Reviewer #3 (Comments to the Authors (Required)):

The manuscript by Wiese et al. examines KANSL3, a core component of the NSL complex, in liver development and homeostasis. By deleting floxed *Kansl3* using Albumin-Cre, the authors observe early-onset liver disease, fibrosis, and biliary hyperplasia, suggesting that KANSL3 is essential for hepatocyte differentiation and liver function. They extend these findings to an in vitro organoid system, where KANSL3 deficiency impairs hepatocyte differentiation. Overall, the data convincingly show that KANSL3-mediated regulation is critical for liver development. However, the statements regarding adult liver plasticity and regeneration remain to be substantiated. If confirmed, this study would advance the understanding of how specific epigenetic regulators maintain hepatic function and plasticity.

We appreciate the reviewer's comments and thank them for acknowledging the significance of our findings to "convincingly show that KANSL3-mediated regulation is critical for liver development".

Major Comments

1. KANSL3's Role in Liver Development vs. Maintenance/Regeneration

The authors provide strong evidence that Albumin-Cre-mediated *Kansl3* deletion leads to severe liver pathology (fibrosis, biliary hyperplasia) by three weeks of age. The results from serum markers and Picrosirius red staining support a pronounced disease phenotype. However, page 12 states that "deleting KANSL3 in 1-week-old mice leads to early-onset liver disease," even though Albumin-Cre is active much earlier, so the data primarily implicate KANSL3 in fetal or neonatal development rather than strictly adult maintenance.

We agree with the reviewer that our previous statement was misleading. The AlbCre transgene is active from E15.5, and our data primarily implicate KANSL3 in fetal or neonatal development rather than adult maintenance. Interrogating the role of KANSL3 at different stages of liver development and in adult tissue maintenance in vivo would require an inducible mouse model which we believe is beyond the scope of the current manuscript. We have updated the text accordingly.

However, our results suggest that KANSL3 function is crucial for hepatocyte differentiation early in fetal liver development. To test this we generated fetal liver organoids from *Kansl3^{fl/fl}* and

Cag-CreERT2T/+;Kansl3fl/fl E12.5 embryos and observed growth defects, reduced Cyp3a4 activity (Fig. 6D), and impaired Albumin secretion (Fig. 6E) upon Kansl3 knockout. These findings support the notion that KANSL3 is essential for hepatocyte differentiation and function during fetal development.

We have revised our statements to focus on the establishment of the phenotype due to changes early in liver development, rather than regeneration and liver plasticity. Our results emphasize KANSL3's critical role in hepatocyte differentiation and function, both during BEC-to-Hep transdifferentiation *in vivo* and *in vitro* and hepatoblast differentiation into hepatocytes *in vitro*. However, further work is needed to understand the exact molecular mechanisms.

Neonatal murine cells sorted for EPCAM/TSPAN8/CD24 are used in organoid assays to model a "late" Kansl3 deletion, but these markers also label oval cells that can give rise to both hepatocytes and cholangiocytes. This assay, therefore, does not exclusively distinguish plasticity from delayed or abnormal differentiation. It remains unclear whether the observed defects *in vivo* and *in vitro* reflect a genuine shift in cell fate (plasticity) or simply impaired/delayed maturation.

We appreciate the reviewer's concern that our organoid assay may not exclusively distinguish plasticity from delayed or abnormal differentiation. Indeed, EPCAM high/TSPAN8 high/CD24 positive cells were used in organoid assays to model differentiation of bipotent BEC progenitors or "oval cells" into hepatocytes as suggested by our scRNA-seq analysis. However, given the mouse phenotype we observed in an Alb-Cre driven KO mouse model, our data suggests that KANSL3 is involved in hepatocyte differentiation and/or maturation, rather than a shift in cell types from hepatocytes to BECs, albeit we cannot rule it out completely.

Our *in vitro* results show that essential gene expression programs for hepatocyte differentiation are not induced without KANSL3 in BEC-to-Hep organoids. We hypothesize that BEC progenitor organoids may persist in a more BEC-like state in the absence of KANSL3, rather than undergoing a complete shift towards mature BECs. This is supported by the fact that we did not observe an increase in BEC markers during Hep-like differentiation of fetal liver organoids upon deletion of KANSL3 (see data below, significance was scored by Mann-Whitney test).

Notably, we observed reduced proliferation of organoids upon KANSL3 deletion (Fig. 5C), indicating that impaired hepatocyte differentiation likely arises from a complex interplay of defects in hepatic metabolic gene expression and broader cellular processes.

Upon revision, we also showed that *Kansl3* LKO hepatocytes in neonates undergo increased apoptosis (Fig. 3I), which likely contributes to an imbalance between hepatocytes and BECs.

We conclude that our results are more in line with delayed hepatocyte differentiation/maturation defects rather than a change in cell types, which is paired with defects in broader cellular processes such as proliferation and viability. We have revised the text and discussion to acknowledge the limitations of our models.

If the manuscript intends to emphasize a role for KANSL3 in adult hepatocyte maintenance or regeneration, data from inducible deletion in mature hepatocytes (e.g., via tamoxifen-regulated Cre) coupled with adult liver injury or regeneration experiments would clarify whether KANSL3 deficiency disrupts genuine adult plasticity rather than a developmental process. Otherwise, claims regarding adult roles must be adjusted.

We thank the reviewer for this critical evaluation. As stated above, claims regarding regeneration or plasticity have been toned down.

2. "Unprecedented Role in Adult Liver Regeneration"

The data convincingly demonstrate KANSL3's importance for liver development and homeostasis, but stating that it provides a "unique mechanism" or "unprecedented role in mediating regeneration of the adult liver" appears overstated. Another NSL subunit, MCRS1, has already been shown to influence adult hepatocyte biology and pathological outcomes (PMID: 34958836). Furthermore, the presented data focus largely on neonates, rather than adult animals.

This point could be addressed by clarifying that KANSL3 contributes to hepatic maintenance and development. If the authors wish to underscore a role in adult regeneration, additional evidence such as adult-onset deletion or injury-based models would be required. Mention of MCRS1 is also relevant because Figure 1B shows that KANSL3 depletion reduces MCRS1, suggesting a broader NSL complex function in liver health.

We thank the reviewer for their critical evaluation of our manuscript. As previously mentioned, we have revised our claims regarding adult regeneration and plasticity, and have removed or toned down these statements to ensure that our findings are accurately represented. Additionally, we have added a new section to the discussion, where we consider our data in the context of existing reports on the function of other NSL complex members in liver health.

3. Single-Cell Analysis

The scRNA-seq data convincingly indicate an imbalance between hepatocytes and biliary epithelial cells in neonatal knockout livers, reflecting impaired hepatocyte maturation. However, deducing that KANSL3 is essential for adult plasticity remains challenging without adult deletion or injury-based lineage tracing.

We thank the reviewer for this comment. As mentioned above, we agree that based on the data presented in the manuscript we cannot claim a role for KANSL3 in adult liver plasticity. We agree with the reviewers on the importance of employing an inducible Alb-Cre driven KO model coupled with lineage tracing to study KANSL3's role in liver regeneration. Unfortunately, developing new mouse models and obtaining the necessary animal ethics license in Germany is a lengthy process, exceeding the current revision timeline. For this reason, we have removed our claims regarding KANSL3's role in regulating adult liver plasticity.

It would be helpful to present KANSL3 expression in different clusters (C1-15) or along the proposed differentiation trajectory (as in Fig. S4E) in order to show how KANSL3 (and possibly other NSL components) is regulated under normal or stressed conditions.

We have provided the expression of NSL complex members across the differentiation trajectory as requested by the reviewer. However, due to the overall low expression of these genes, we are unable to make any meaningful claims about a potential link between NSL mRNA levels and the differentiation trajectory, although NSL members appear to be slightly more expressed in the BEC compartment.

Similarly, we have examined the correlation between KANSL3 expression and the determination of hepatocyte cell fate, but found no significant correlation, similar to Sox9 (Fig. 4G, bottom). This is in contrast to the clear trend observed in the correlation between Alb expression (Fig. 4G, top) and hepatocyte fate bias as expected.

Clusters 12, 13, and 15 also exhibit marked differences between genotypes yet receive minimal discussion.

Apart from cluster 12, the total number of cells in cluster 13 and 15 is below 100 cells. Given this low number, we believe that the results obtained from this analysis may not be significant enough to discuss further in the manuscript. However, upon revision we have updated our annotation of the stromal cell clusters C3, C8, C10, C11 and C12 (Fig. 3D). The data concerning these cell types are now contained in Supplemental Fig. S6A-E. Stromal cells are found in the clusters C3, C8, C10, C11 and C12. For clusters 3 and 10 we observed a reduction of cells in the Kansl3 LKO mice (Supplemental Fig. S6D). Cluster 8 showed a mild increase in Kansl3 LKO mice, while a more pronounced increase in cell number was observed for cluster 12. The numbers of cells within the different stromal cell populations overall do not indicate a significant shift as observed in the epithelial cell populations, however, differential expression analysis between controls and Kansl3 LKO cells combining all stromal cells showed an upregulation of pathways related to fibrosis and tissue reorganization such as collagen catabolism, keratan sulfate degradation or disease of glycosylation (Supplemental Fig. S6E), which is in line with the increased fibrosis observed *in vivo*.

More direct comparisons of differentially expressed genes (DEGs) between mutant and wild-type cells in each cluster, rather than just showing cluster-specific marker genes (as in Fig. 2H-I), could further strengthen the single-cell dataset's interpretation.

We thank the reviewer for their suggestion to further strengthen the interpretation of our single-cell dataset. We have revised our analysis to include pseudobulk differential expression analyses between control and Kansl3 LKO conditions for the hepatocyte cluster C0, the BEC cluster C1, and the stromal cells (C3, C8, C10, and C12 combined). These analyses are presented in Fig. 3H, Supplemental Fig. S6E and Supplemental Table S4.

In the hepatocyte cluster C0, we observed a down-regulation of genes related to mitochondrial function, ciliary landscape, fatty acid metabolism and response to toxic substances (Fig. 3H, left). Conversely, upregulated genes were enriched for apoptosis, ferroptosis, and stress

signaling. These findings support our previous observations of hepatocyte dysfunction in *Kansl3* LKO mice and further highlight the decrease in viability in the hepatocyte compartment.

In the BEC cluster C1, we detected a down-regulation of similar pathways related to mitochondria and lipid metabolism (Fig. 3H, right), which may be due to the heterogeneous nature of BECs and their broad expression of *Alb* (Fig. 3E). However, we also observed an upregulation of cell-cell adhesion, proliferation, and VEGFA signaling, which are consistent with the ductular response observed in these animals.

4. Molecular Mechanism: NSL Complex and Histone Acetylation

The ChIP-seq analysis for *KANSL3* in wild-type livers, overlaid with DEG promoters, suggests direct regulation of down-regulated genes. However, selection criteria (e.g., 490 genes vs. all 3148 down-regulated genes) require clarification.

The $n=3148$ genes represent all down-regulated genes identified with an FDR (p_{adj}) <0.05 , without any fold change cutoff applied. In contrast, the $n=490$ genes include those showing expression changes with $\log_2FC < -1$, representing the most strongly impacted genes by *Kansl3* deletion. For downstream analysis, we focused on genes with $\log_2FC < -1$ for down-regulation and $\log_2FC > 1$ for upregulation. We have clarified these criteria in the text for better understanding.

The NSL complex is known to mediate histone H4 acetylation, so measuring relevant histone marks (H4K16ac, H4K12ac, etc.) after *Kansl3* deletion would offer more mechanistic insight and confirm that NSL-mediated acetylation is altered in specific gene targets in the liver.

To explore *KANSL3*'s role in transcriptional regulation and overcome limitations of bulk tissue analysis, we used the dTAG system in the murine Hepa1-6 cell line to model hepatocyte transcriptomic changes following acute *KANSL3* depletion. RNA-seq analysis performed 24 hours post-degradation revealed significant transcriptomic alterations, with 2,202 genes down-regulated and 1,762 upregulated compared to control treatment (FDR <0.05 , Supplemental Fig. 3B, Table S2). Down-regulated genes were predominantly associated with mitochondrial functions, lipid metabolism, and cilia-related terms (Supplemental Fig. 3C,D), suggesting parallels between acute cellular and long-term liver tissue *KANSL3* depletion. Notably, these expression changes were rescued by ectopic expression of a FLAG-tagged human *KANSL3* full-length protein (Supplemental Fig. 3E-G), confirming direct regulation by *KANSL3*.

KANSL3 ChIP-qPCR confirmed binding specificity to promoters of target genes, such as *Mrp11* and *Fah*, which was abolished upon dTAG-13 treatment (Supplemental Fig. 3H). ChIP-seq analysis demonstrated reduced H4K16ac levels at *KANSL3* target genes following degradation, underscoring histone acetylation-dependent regulation by the NSL complex (Supplemental Fig. 3I). Additionally, treating dTAG-13 degraded cells with the HDAC inhibitor Tricostatin A (TSA) reversed gene expression effects, with *Mrp11* returning to control levels and *Fah*, *Mrp55*, and *Ndof3b* exceeding them (Supplemental Fig. S3J). These findings support an acetylation-dependent regulatory mechanism by *KANSL3* in hepatocyte cell lines. We believe *KANSL3* similarly regulates gene expression in hepatocytes in vivo.

While we aimed to investigate the effect of KANSL3 degradation on other NSL-complex associated histone H4 acetylation marks (H4K5ac and H4K8ac), technical challenges prevented us from doing so within the timeframe of revision. However, given the known preference of NSL depletion for K5 and K8 residues in other cellular systems (PMID: 33657400), we anticipate that the observed changes would be even more pronounced as for H4K16ac.

5. Additional Functional Readouts

The transcriptomic data highlight disruptions in lipid and steroid metabolism. Oil Red O staining at E17.5, P7, and P21 might reveal the timing of any lipid accumulation or liver metabolic dysfunction. Staining for and quantifying canonical hepatocyte markers, such as HNF4A, at multiple stages would further support the conclusions about hepatocyte-to-BEC identity shifts and their timing in development.

We thank the reviewer for their suggestion to further investigate the timing of lipid accumulation and liver metabolic dysfunction in Kansl3 LKO mice. Upon revision, we have added Oil Red O staining in P21 animals, which shows increased lipid deposition in Kansl3 LKO animals (Fig. 2D). We have also measured total cholesterol levels in livers of P7 and P21 mice and found that while levels remained equal at P7, they increased in Kansl3 LKO livers at P21 (Fig. 2E). These findings support our RNA-seq results and suggest that metabolic defects are slowly building up in the livers of neonatal animals. These changes may also be related to the transition to solid food around weaning, which falls into this time window.

Additionally, we generated Hep-like fetal liver organoids and found that loss of Kansl3 affects hepatocyte function (Fig. 6D,E), indicating a role for KANSL3 already early on during hepatic specification in the fetal liver.

As previously discussed, we have toned down our claims on a potential hepatocyte-to-BEC identity shift and instead conclude that our results are more in line with delayed hepatocyte differentiation/maturity defects rather than a change in cell types, which is paired with defects in broader cellular processes such as proliferation and viability.

We have done our best to address the reviewers' concerns within the time constraints of this revision. We have revised the text and discussion to acknowledge the limitations of our new data and provide a clear understanding of our results.

6. Early or Transient Cell Death?

The text indicates that apoptosis is not elevated at P7, suggesting that cell death is unlikely to fully explain the altered cell compositions. However, cell death could have occurred at an earlier stage. More comprehensive time-course analyses or additional histological measures of apoptosis might help corroborate whether the effect on hepatocyte numbers is indeed due to differentiation defects rather than early or transient cell death.

We appreciate the reviewer's focus on apoptosis and have addressed this extensively. We enhanced our scRNA-seq analysis by conducting pseudobulk differential expression analysis

between control and *Kansl3* LKO conditions for hepatocyte cluster C0. GO analyses revealed significant upregulation of apoptosis and ferroptosis-related pathways (Fig. 3H), contrasting our earlier submission, which examined only selected pro-apoptotic genes.

To corroborate these findings, we developed a new mouse model by crossing *Kansl3* LKO and AlbCre controls with mTmG reporter mice to genetically label Alb-expressing cells. Single-cell suspensions from P7 livers were stained with EpCAM antibodies to separate hepatocytes from BECs and intermediate cells expressing both markers (Fig. 3H). This analysis confirmed the observed imbalance between hepatocytes and BECs in P7 mTmG *Kansl3* LKO mice compared to controls (Supplemental Fig. S5N-P). Furthermore, Annexin V staining showed increased apoptosis of hepatocytes in P7 mTmG *Kansl3* LKO mice versus controls (Fig. 3H/I), supporting our gene expression findings.

Though we don't consider apoptosis the sole factor in the hepatocyte-BEC imbalance, our results indicate that it contributes significantly.

Minor Comments

- Figure 2F/G: A clearer labeling of yellow vs grey clusters (e.g., highlighting which cells are knockout vs. control) would improve readability (and is nicely done in the other figures).

Updated.

- Abbreviations: Spell out "NSL complex" and similar terms upon first mention.

Updated.

- Figure 1B: Include quantitative Western blot data rather than relying solely on representative images.

Fig. 1B displays Western blot results from three individual mice, all demonstrating a consistent trend upon depletion of *KANSL3*. We respectfully disagree with the reviewer regarding the necessity of quantifying these images, as the representative blots already sufficiently illustrate the findings and replicate the observed pattern in independent mouse tissues.

- Collagen/ α -SMA: Include quantifications for fibrotic markers (e.g., Figure S1G,H).

We thank the reviewer for this comment. We have repeated the experiment and quantified both the area of α -SMA and COLLAGEN staining, both showing increased staining in *Kansl3* LKO samples (Fig. 1I/J). In addition, we performed co-staining of both α -SMA and COLLAGEN to qualitatively show the expected overlap of both proteins (Supplemental Fig. S1G).

- References to Cholangiocyte-Derived Hepatocyte Regeneration: The Forbes lab (PMID: 28700576) showed that cholangiocyte contribution to new hepatocytes requires significant hepatocyte impairment, implying that such BEC-to-hepatocyte transitions do not occur under

normal conditions. The discussion might acknowledge that plasticity typically requires acute or chronic injury with a block in hepatocyte-mediated regeneration.

We thank the reviewer for this suggestion. We have updated the discussion.

September 5, 2025

RE: Life Science Alliance Manuscript #LSA-2025-03238-TR

Dr. Asifa Akhtar
Max Planck Institute of Immunobiology and Epigenetics
Stübeweg 51
Freiburg 79108
Germany

Dear Dr. Akhtar,

Thank you for submitting your revised manuscript entitled "KANSL3 directs transcriptional programmes essential for hepatic metabolism and differentiation". Your manuscript was evaluated by two of the original reviewers who commend the improvement in this revised version.

We would be happy to publish your paper in Life Science Alliance pending final revisions necessary to meet our formatting guidelines.

- Please tend to the minor point from Reviewer 1 related to a potential repetition.
- Please include list of primers to the section on materials used in the manuscript. Please format Table S7 to fit contents into one page width.
- In the methods section: for microscopy related descriptions, please include information on objectives used (magnification, N.A. specified).
- Figure S9 has only one panel, hence, please remove A from the figure, its legend, and call-out in the manuscript text.
- Please upload a clean manuscript file without colored text.
- Please consult our manuscript preparation guidelines <https://www.life-science-alliance.org/manuscript-prep> and make sure your manuscript sections are in the correct order.
- Please add your main, supplementary figure, and table legends to the main manuscript text after the references section.
- Please upload all figure files as individual ones, including the supplementary figure files; all figure legends should only appear in the main manuscript file.
- Please add the X and Bluesky handles of your host institute/organization, as well as your own and/or one of the authors, in our system.
- please be sure that the authorship listing and order is correct.

A. FINAL FILES:

-- Summary blurb (enter in submission system): A short text summarizing in a single sentence the study (max. 200 characters including spaces). This text is used in conjunction with the titles of papers, hence should be informative and complementary to

the title. It should describe the context and significance of the findings for a general readership; it should be written in the present tense and refer to the work in the third person. Author names should not be mentioned.

B. MANUSCRIPT ORGANIZATION AND FORMATTING:

Thank you for your attention to these final processing requirements. Please revise and format the manuscript and upload materials as soon as you are able.

Sincerely,

Sarita Hebbar, PhD
Scientific Editor
Life Science Alliance
<http://www.lsjournal.org>

Reviewer #1 (Comments to the Authors (Required)):

I congratulate the authors for their changes to the paper, especially for the new data strengthening the organoid assays (including additional organoid model), long term analysis of the KANSL3-KO phenotype, functional assays and clarification of which metabolism genes are affected by the KO, clarification on the assignment of mesenchyme clusters, and providing new data of Alb-Cre mTmG mice more clearly showing the imbalance of hepatocytes and cholangiocytes seen in the KO. Some limitations of the study remain, but the authors clearly state them in text and discussion. I am happy to recommend this revised manuscript for publication.

I have detected a small direct text repetition, which should be corrected in the proofs stage:

"Our studies revealed that the deletion of KANSL3 in hepatocytes in vivo leads to 79 early-onset liver disease, characterized by biliary hyperplasia and fibrosis. We found that the 80 deletion of KANSL3 in hepatocytes in vivo leads to early-onset liver disease, characterized 81 by biliary hyperplasia and fibrosis."

Reviewer #3 (Comments to the Authors (Required)):

The authors have addressed all my concerns. I congratulate the authors on the nice work and recommend publication.

Point-by-point response to editorial request, our responses are marked in blue:

1. Please address the minor point from Reviewer 1 regarding a potential repetition.
The repetition has been removed in the final version of the manuscript.
2. Please include the list of primers in the Materials Used section of the manuscript and reformat Table S7 so that its contents fit within one page width.
We had assumed that Table S7 could be submitted in Excel format with multiple tabs, but it appears this is not acceptable. We have now consolidated all tabs into a single Excel sheet. Unfortunately, fitting all the information onto a single page would significantly compromise readability. Please let us know if the current format is acceptable, or if you would prefer that we adjust the table further to meet the formatting requirements.
3. In the Methods section, for microscopy-related descriptions, please include information on the objectives used (magnification and numerical aperture specified).
In the final manuscript, we had included the magnification of the objectives used to acquire the images, but we inadvertently omitted the numerical aperture. This information has now been added to the Methods section in the revised manuscript, which is attached to this email.

September 16, 2025

RE: Life Science Alliance Manuscript #LSA-2025-03238-TRR

Dr. Asifa Akhtar
Max Planck Institute of Immunobiology and Epigenetics
Stübeweg 51
Freiburg 79108
Germany

Dear Dr. Akhtar,

Thank you for submitting your Research Article entitled "KANSL3 directs transcriptional programmes essential for hepatic metabolism and differentiation". It is a pleasure to let you know that your manuscript is now accepted for publication in Life Science Alliance. Congratulations on this interesting work.

DISTRIBUTION OF MATERIALS:

Again, congratulations on a very nice paper. I hope you found the review process to be constructive and are pleased with how the manuscript was handled editorially. We look forward to future exciting submissions from your lab.

Sincerely,

Sarita Hebbar, PhD
Scientific Editor
Life Science Alliance
<http://www.lsajournal.org>